# Phosphoproteomic screening identifies physiological substrates of the CDKL5 kinase

Ivan M Muñoz[1,†], Michael E Morgan[1,†], Julien Peltier[1,2], Florian Weiland[1], Mateusz Gregorczyk[1], Fiona CM Brown[1], Thomas Macartney[1], Rachel Toth[1], Matthias Trost[1,2,*] & John Rouse[1,**]

## Abstract

Mutations in the gene encoding the protein kinase CDKL5 cause a debilitating neurodevelopmental disease termed CDKL5 disorder. The impact of these mutations on CDKL5 function is poorly understood because the substrates and cellular processes controlled by CDKL5 are unclear. Here, we describe a quantitative phosphoproteomic screening which identified MAP1S, CEP131 and DLG5—regulators of microtubule and centrosome function—as cellular substrates of CDKL5. Antibodies against MAP1S phospho-Ser[900] and CEP131 phospho-Ser[35] confirmed CDKL5-dependent phosphorylation of these targets in human cells. The phospho-acceptor serine residues in MAP1S, CEP131 and DLG5 lie in the motif RPXSA, although CDKL5 can tolerate residues other than Ala immediately C-terminal to the phospho-acceptor serine. We provide insight into the control of CDKL5 activity and show that pathogenic mutations in CDKL5 cause a major reduction in CDKL5 activity *in vitro* and in cells. These data reveal the first cellular substrates of CDKL5, which may represent important biomarkers in the diagnosis and treatment of CDKL5 disorder, and illuminate the functions of this poorly characterized kinase.

**Keywords** CDKL5 disorder; centrosome; cilia; kinase; microtubule
**Subject Categories** Cell Adhesion, Polarity & Cytoskeleton; Genetics, Gene Therapy & Genetic Disease; Post-translational Modifications, Proteolysis & Proteomics
The EMBO Journal (2018) 37: e99559

See also: **LL Baltussen et al** (December 2018) and **PA Eyers** (December 2018)

## Introduction

Cyclin-dependent kinase-like 5 (CDKL5/STK9) is a poorly understood protein kinase mutated in human disease. *CDKL5* mutations were first described in 2003 in two girls affected by infantile spasms (Kalscheuer *et al*, 2003). Some of the symptoms associated with

*CDKL5* mutations overlap with Rett syndrome (RTT) caused by mutations in *MECP2*, which has resulted in some of the patients with *CDKL5* mutations being classified as having an early-onset seizure variant of RTT (ESV-RTT). Other patients with mutations in *CDKL5* have been variably classified as having early infantile epileptic encephalopathy, X-linked dominant infantile spasm syndrome or diagnosed with other epileptic conditions (Tao *et al*, 2004). Now, however, a distinct condition referred to as CDKL5 disorder has been recognized as an independent clinical entity, with early-onset epilepsy and severe neurodevelopmental delays as defining hallmarks. In particular, CDKL5 disorder is characterized by seizure onset before 3 months of age, severely impaired gross motor, language and hand skills and subtle but shared physical characteristics (Fehr *et al*, 2013). Over 800 cases of CDKL5 disorder have been reported worldwide, and this number is increasing as awareness of CDKL5 disorder grows (Krishnaraj *et al*, 2017).

The *CDKL5* gene is found on the X chromosome, and several transcript isoforms have been reported although the major isoform appears to be 1,030 amino acid long (CDKL5[115]; Hector *et al*, 2016; Fig 1A). The vast majority of patients with CDKL5 disorder are female, with a spectrum of severities that is most likely related to the pattern of X-inactivation, which is known to be random. The condition is more severe in male than in female presumably because there is no wild-type CDKL5 in affected individuals (Fehr *et al*, 2013). The CDKL5 gene product belongs to the CMGC branch of the human kinome and more specifically to a sub-branch comprising kinases CDKL1, 2, 3, 4 and 5 (Manning *et al*, 2002). These kinases are so-called because of similarity in the kinase domain to cyclin-dependent kinases (CDKs) although they are also highly similar to MAP kinases (MAPKs; data not shown). Phosphorylation of MAPKs and CDKs in the well-characterized activation T-loop, a variable region between catalytic subdomains VII and VIII, is a common mechanism activating these kinases (Cobb & Goldsmith, 1995; Roux & Blenis, 2004; Lolli & Johnson, 2005). In some kinases, T-loop phosphorylation is achieved by autophosphorylation, but often, it is catalysed by upstream kinases. For example, the classical MAPKs termed ERK1 and ERK2 are inactive before MAP kinase kinase 1-catalysed phosphorylation of Thr[183] and Tyr[185] in each T-loop; these residues lie in a TEY motif, and phosphorylation of both residues is

1 MRC Protein Phosphorylation and Ubiquitylation Unit, School of Life Sciences, University of Dundee, Dundee, UK
2 Faculty of Medical Sciences, Institute for Cell and Molecular Biosciences, Newcastle upon Tyne, UK
*Corresponding author. Tel: +44 191 2087009; E-mail: matthias.trost@ncl.ac.uk
**Corresponding author. Tel: +44 1382 385490; E-mail: j.rouse@dundee.ac.uk
†These authors contributed equally to this work

required for ERK1 activity (Anderson *et al*, 1990; Cobb & Goldsmith, 1995; Roux & Blenis, 2004). Human CDKL5 has a TEY motif in the T-loop, and it has been proposed that this motif in CDKL5 is recognized by phospho-specific antibodies raised against the ERK1 TEY motif, although it was not demonstrated that cross-reactivity with CDKL5 requires an intact TEY motif (Lin *et al*, 2005; Bertani *et al*, 2006). The sites of phosphorylation and autophosphorylation on CDKL5 in cells have not yet been mapped, and the mechanisms controlling kinase activity are unknown.

Although a range of proteins have been described as potential substrates of CDKL5, the proteins phosphorylated by CDKL5 in cells remain to be identified, and no systematic screen for cellular CDKL5 substrates has been reported. Identification of the key substrates of CDKL5 is vital, as this information would illuminate the major cellular functions of this kinase and may provide biomarkers that would aid the diagnosis and treatment of CDKL5 disorder. Being able to measure CDKL5 activity in cells would also allow unambiguous assessment of the functional impact of pathogenic mutations on CDKL5, most of which map to the kinase catalytic domain (Fig 1A). In this study, we report the results of a global phosphoproteomic screening for CDKL5 substrates, which revealed the first physiological substrates of CDKL5 that in turn has allowed us to test the effect of pathological mutations on CDKL5 function, and enabled us to obtain the first clues as to how CDKL5 activity is controlled in cells.

## Results

### TMT-based CDKL5 phosphoproteomic screening

To identify CDKL5 substrates, we undertook a quantitative phosphoproteomic screening, which compared the total phosphoproteomes of *CDKL5* knockout (KO) cells and *CDKL5* KO cells in which CDKL5 was stably re-expressed. We chose this strategy to avoid clonal differences between knockout cells and parental cells. First, CRISPR-/Cas9-mediated genome editing was used to disrupt the *CDKL5* gene in U2OS osteosarcoma cells modified with the Flp-In™ T-REx™ system. Around 35 clones were screened for CDKL5 loss by Western blotting using in-house CDKL5 antibodies (data not shown); two of the knockout clones (clones 7 and 13) are shown in Fig 1B. Genomic sequencing and RT–PCR revealed that clone 7 had no wild-type CDKL5 allele (data not shown), but instead, two

different classes of disrupted *CDKL5* allele were identified that result in truncation at amino acids 62 and 75, respectively (Appendix Fig S1A and B). No wild-type CDKL5 allele was detected in clone 13 (data not shown), and instead, a single class of disrupted *CDKL5* allele was identified bearing a mutation that truncates the protein product at residue Val[38] (Appendix Fig S1C and D). The Flp-In™ T-REx™ system allows stable, tetracycline (Tet)-inducible expression of a gene of interest from a specific genomic location. The *CDKL5* open reading frame was introduced at the FRT sites of *CDKL5* knockout (KO) clone 13. Incubation of these cells with Tet allowed stable expression of CDKL5 (Fig 1C; data not shown).

Three biological replicates of *CDKL5* KO cells stably transfected with empty vector or *CDKL5* KO cells stably expressing CDKL5 were lysed, cysteines were reduced and alkylated, and protein extracts were digested with trypsin. Phosphopeptides were enriched using titanium dioxide ($TiO_2$) chromatography, and the six samples were isotopically labelled using tandem mass tags (TMT; Fig 1D). TMT labelling allowed quantitative, multiplexed analysis of all six samples, which were combined and analysed in a parallel manner that enables quantitation of tryptic peptides in each of the individual samples with high accuracy (Rauniyar & Yates, 2014). Labelling efficiency was checked in each sample and found to exceed 95%. Samples were pooled and fractionated by basic pH reversed-phase chromatography into 40 fractions which were concatenated into 12 fractions. These fractions were analysed by LC-MS/MS on a high-resolution Orbitrap Fusion mass spectrometer (Fig 1D). A total of 12,316 unique phosphosites were quantified in all replicates at a false-discovery rate (FDR) of < 1%. Data showed high levels of reproducibility between replicates (Appendix Fig S2A and B). A total of 194 phosphosites were higher in abundance in CDKL5-expressing cells compared with CDKL5 KO cells (> 1.5-fold, $P < 0.05$), representing potential CDKL5 substrates, whereas 159 were lower in abundance (Fig 2A; Dataset EV1). Only 29 of the 194 phosphosites that were higher in CDKL5-expressing cells compared with CDKL5 KO cells were threonine-phosphorylated, all on Thr-Pro (TP) motifs (Appendix Fig S3A), whereas the majority (165) were serine-phosphorylated. The phosphosites changing the most were sites of CDKL5 itself: Ser[306], Ser[308], Ser[407], Ser[529] and Ser[543] (Fig 2A; Dataset EV1). Even though these sites lie in Ser-Pro (SP) motifs typically phosphorylated by CDKs and MAPKs, the most highly enriched phospho-motif in the mass spectrometry dataset was RXXS (5.11-fold increase; Appendix Fig S3B). This motif was found in the

**Figure 1. A quantitative phosphoproteomic approach for the identification of cellular CDKL5 targets.**

A  Pathogenic and non-pathogenic CDKL5 variants. Schematic diagram shows modular domains in CDKL5 and the position of amino acid substitutions in humans that are either pathogenic, non-pathogenic or of unknown consequence according to RettBASE (http://mecp2.chw.edu.au/cdkl5/cdkl5_variant_list_copy.php; Krishnaraj *et al*, 2017). NES: nuclear export signal; NLS: nuclear localization signal.

B  Generation of *CDKL5* knockout human cells. U2OS cells modified with the Flp-In™ T-REx™ system were subjected to genome editing to disrupt *CDKL5*. Extracts of cells from two different knockout (KO) clones (7 and 13) were subjected to SDS–PAGE on the gel types indicated followed by immunoblotting with in-house anti-CDKL5 antibodies. "Hi" higher exposure; "lo" lower exposure. Asterisk: non-specific band.

C  Stable expression of CDKL5 in knockout clone 13. The *CDKL5* open reading frame (1,030 amino acid/115-kDa isoform) was inserted at the FRT sites in CDKL5 knockout clone 13 from (B). Cells transfected with empty vector were used as control. Cells were incubated with the indicated concentrations of tetracycline (Tet), and extracts were immunoblotted with anti-CDKL5 antibodies.

D  Phosphoproteomics workflow. *CDKL5* knockout clone 13 and the same cells re-expressing CDKL5 were lysed and protein extracts were digested using trypsin. After phosphopeptide enrichment by $TiO_2$ chromatography, peptides were isotopically labelled by TMT and combined. Combined peptides were fractionated by high-pH reversed-phase chromatography. Fractions were separated on a nano-HPLC and analysed by quantitative mass spectrometry on an Orbitrap Fusion mass spectrometer. Data were analysed using MaxQuant software.

Source data are available online for this figure.

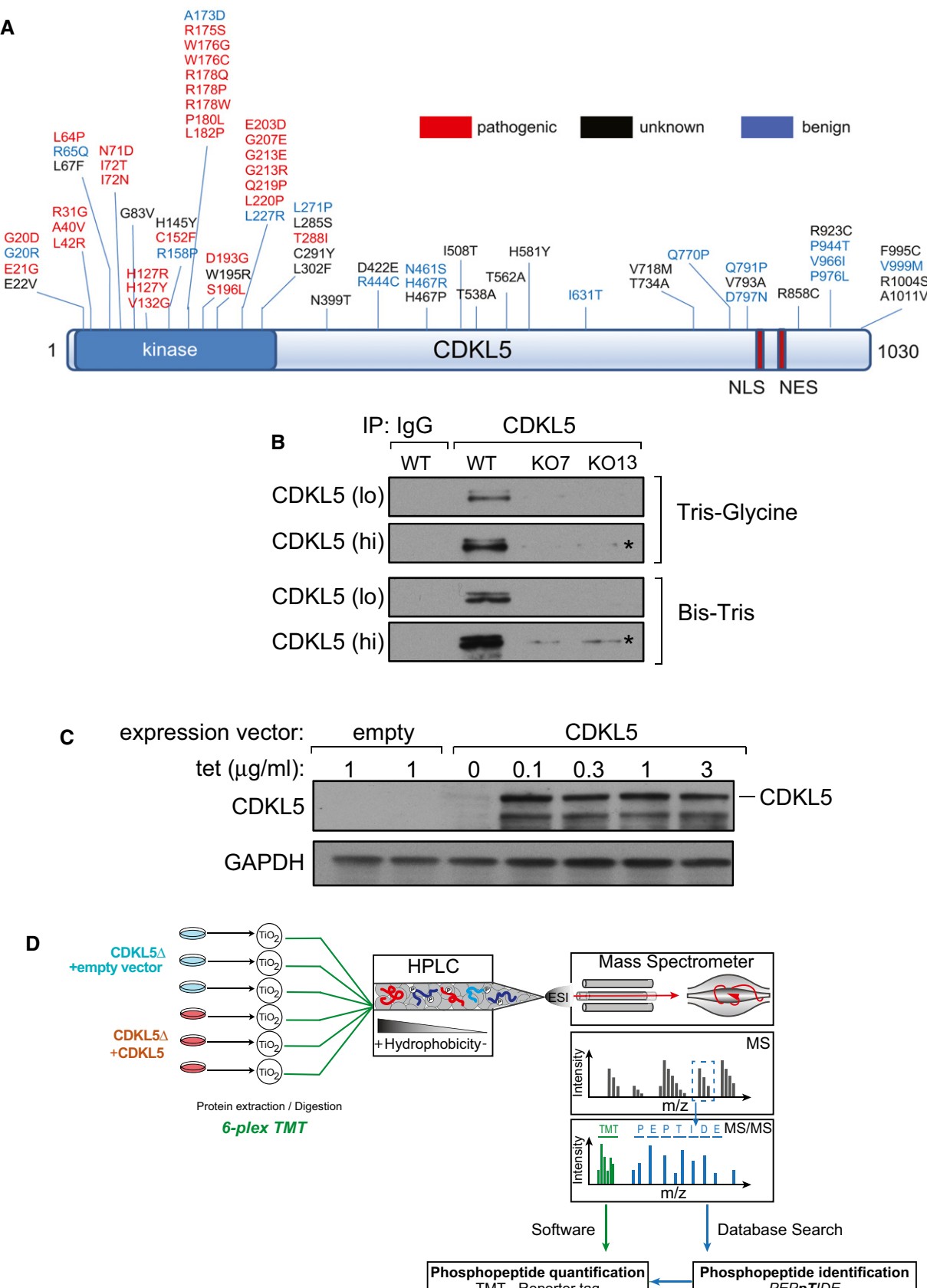

**Figure 1.**

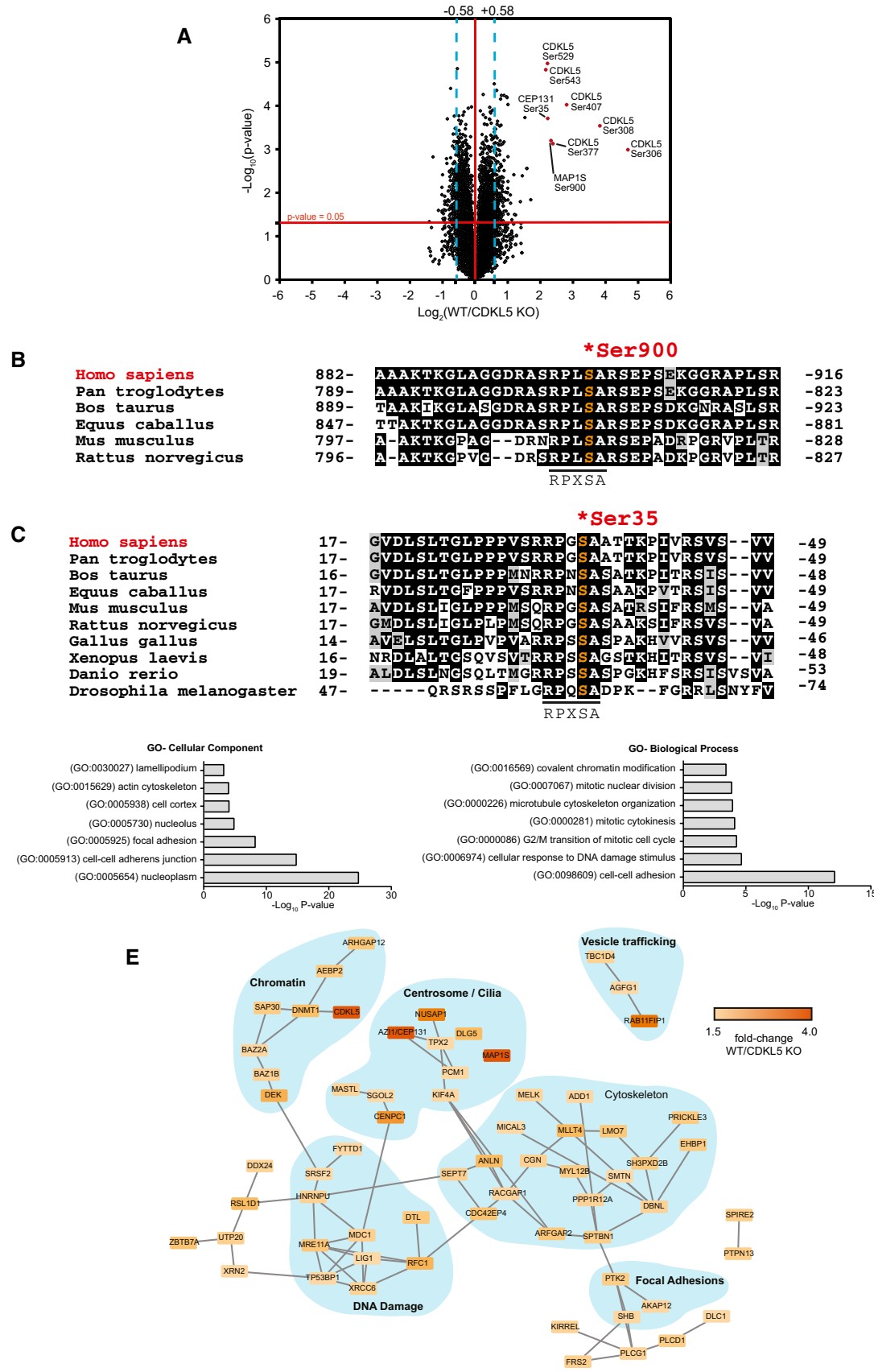

**Figure 2.**

**Figure 2. Identification of CDKL5 substrates in human cells.**

A    Volcano plot showing potential CDKL5 substrates. The horizontal cut-off line represents a *P*-value of 0.05, and the vertical cut-off lines represent a $\log_2$ ratio of 0.58 (~1.5-fold) above which peptides were considered to differ significantly in abundance between CDKL5 KO cells expressing empty vector and CDKL5 KO cells expressing CDKL5. The mass spectrometry proteomics data for this figure have been deposited to the ProteomeXchange Consortium via the PRIDE partner repository (Jarnuczak & Vizcaino, 2017) with the dataset identifier PXD009374.

B, C  Sequence alignment of the Ser[900] phosphorylation site in MAP1S (B) and the Ser[35] phosphorylation site in CEP131 (C), in the species indicated.

D    Gene Ontology (GO) term enrichment of phosphopeptides that are more abundant in CDKL5-expressing cells compared to knockout cells.

E    String database network analysis of proteins which harbour phosphosites up-regulated in CDKL5-expressing cells compared to knockout cells. These are part of protein complexes involved in a wide range of biological functions.

second and third most highly changing hits from the screen: microtubule-associated protein 1S (MAP1S) Ser[900] and CEP131/AZI1 Ser[35], respectively (Fig 2B and C). MAP1S Ser[900] and CEP131/AZI1 Ser[35] both show a high degree of evolutionary conservation among their respective orthologues (Fig 2B and C). Furthermore, the similarity in sequence around the two phosphorylation sites (MAP1S: RPLS[900]A and CEP131: RPGS[35]A) raised the possibility that CKDL5 may favour serine residues in the sequence RPXSA.

Gene Ontology (GO) analysis showed that the proteins less phosphorylated in the absence of CDKL5 were significantly enriched in protein groups involved in DNA damage response, the actin cytoskeleton and microtubule-associated proteins (Fig 2D and E). Interestingly, among these were a number of known proteins involved in the control of primary cilia—cell surface appendages nucleated by centrosomes and structured by microtubules (Basten & Giles, 2013; Fry *et al*, 2014; Canning *et al*, 2018), including MAP1S, CEP131 and DLG5 (Orban-Nemeth *et al*, 2005; Tegha-Dunghu *et al*, 2014). Proteins involved in centromere function such as centromere protein C (CENPC, Ser[146]) and proteins involved in cytokinesis functions such as nucleolar and spindle-associated protein 1 (NUSAP1, Ser[349]/Ser[352]) were also identified. Taken together, these data suggest that CDKL5 directly or indirectly controls the cell cytoskeleton and/or cilium function, consistent with a recent report that overexpression of CDKL5 reduces primary cilium length in human cells (Canning *et al*, 2018).

### Validating CDKL5-dependent phosphorylation of MAP1S and CEP131

In order to test CDKL5-dependent phosphorylation of MAP1S in cells, antibodies were raised in sheep against a phosphopeptide corresponding to the sequence around Ser[900]. Dot blot analysis revealed that affinity-purified antibodies from the second and third bleeds recognized the phosphopeptide immunogen but hardly recognized the non-phosphopeptide equivalent (Fig EV1A). We next tested whether CDKL5 can phosphorylate MAP1S in cells. To this end, MAP1S was co-expressed with CDKL5 in HEK293 cells, and the phosphorylation of MAP1S in anti-FLAG precipitates was examined using the phospho-Ser[900] antibodies. As shown in Fig 3A, expression of wild-type CDKL5 but not a K[42]R kinase-dead CDKL5 mutant markedly increased the phosphorylation of MAP1S. A MAP1S Ser[900]Ala mutation prevented cross-reactivity with the phospho-Ser[900] antibodies (Fig 3A), confirming the specificity. Taken together, these data validate MAP1S as a cellular substrate of CDKL5. A similar analysis was carried out to test CDKL5-dependent phosphorylation of CEP131. Affinity-purified antibodies, raised in sheep against a phosphopeptide corresponding to the sequence around Ser[35], recognized the phosphopeptide immunogen but hardly recognized the non-phosphopeptide equivalent (Fig EV1B).

As shown in Fig 3B, expression of wild-type CDKL5 but not a Lys[42]Arg (K[42]R) kinase-dead CDKL5 mutant markedly increased the phosphorylation of FLAG-CEP131. The CEP131 Ser[35]Ala mutation prevented cross-reactivity with the phospho-Ser[35] antibodies. Taken together, these data validate CEP131 as a cellular substrate of CDKL5.

As CDKL5 belongs to a family of five related kinases (Fig EV2A), we tested the ability of the CDKL5-related kinases CDKL1, 2, 3 and 4 to phosphorylate MAP1S and CEP131. We found that CDKL1 and CDKL2 only drove very weak phosphorylation of MAP1S in HEK293 cells, and CDKL1 only drove very weak phosphorylation of CEP131 even though the level of expression of CDKL1 and 2 was considerably higher than CDKL5 (Fig EV2B and C). It is worth pointing out that CDKL1–4 may simply be inactive—or minimally active—under the conditions used in this study, but perhaps their activity (and perhaps the activity of all CDKLs) is regulated in response to environmental cues or extracellular or intracellular stimuli.

### Towards a CDKL5 consensus sequence

We wished to test whether CDKL5 can phosphorylate MAP1S Ser[900] and CEP131 Ser[35] directly and also to investigate the sequence determinants that influence phosphorylation of these serine residues by CDKL5. To this end, peptides were synthesized corresponding to the sequence around MAP1S Ser[900] and CEP131 Ser[35], and also around the phosphorylation sites in PDZD4 (pSer[230]), RASSF7 (pSer[159]), AFADIN (pSer[217]), FERMT2 (pSer[192]) and NUSAP1 (pThr[355]) which were also among the top hits in the phosphoproteomic screening (Fig 4A). Two lysine residues were added at the N-terminus of each peptide to enable binding to P81 phosphocellulose paper, which enabled isolation of peptides at the end of kinase reactions and quantitation of peptide phosphorylation (see Materials and Methods section). As shown in Fig 4B, the MAP1S Ser[900] peptide was efficiently phosphorylated by FLAG precipitates from extracts of cells expressing C-terminally FLAG-tagged CDKL5 but not the FLAG-CDKL5 K[42]R kinase-dead mutant. Robust phosphorylation of the CEP131 Ser[35] peptide was also observed, but none of the other substrate peptides were phosphorylated (Fig 4B). These data indicate that CDKL5 can phosphorylate Ser[900] of MAP1S and Ser[35] of CEP131 directly. We also tested the ability of the CDKL5-related kinases CDKL1, 2, 3 and 4 to phosphorylate the MAP1S Ser[900]-containing peptide. As shown in Fig EV2D, FLAG-CDKL2 (but not FLAG-CDKL1, 3 or 4) immunoprecipitated from cells phosphorylated the MAP1S peptide at around 50% of the efficiency observed for CDKL5, although CDKL2 expression was significantly higher than CDKL5.

It is interesting to note that MAP1S Ser[900] and CEP131 Ser[35] both lie in an Arg-Pro-X-Ser-Ala (RPXSA) motif (Fig 2B and C), whereas the serine residues in the other putative substrates which were not phosphorylated directly by CDKL5—PDZD4 (pSer[230]), RASSF7

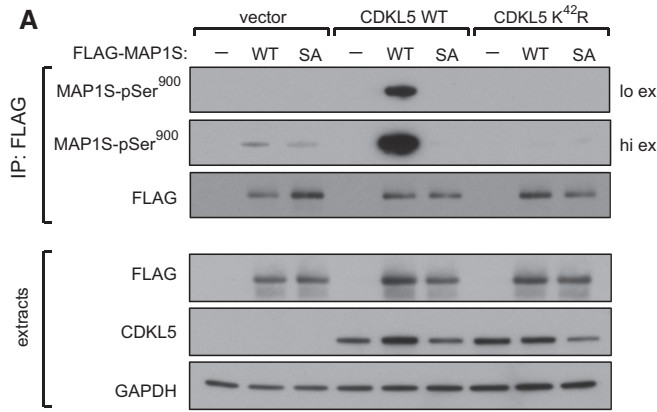

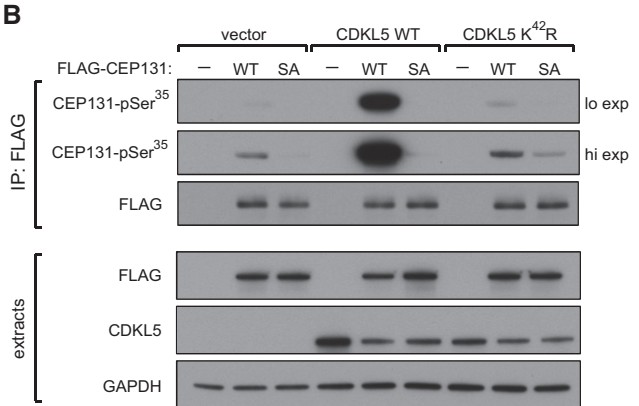

**Figure 3. CDKL5-dependent phosphorylation of MAP1S and CEP131 in cells.**

A  CDKL5 phosphorylates MAP1S at Ser[900] in human cells. HEK293 cells were co-transfected with CDKL5 (wild type "WT" or kinase-dead K[42]R mutant) and/or FLAG-MAP1S (wild type "WT" or a S[900]A mutant "SA"). Anti-FLAG precipitates were subjected to Western blotting with the antibodies indicated. "Hi" higher exposure; "lo" lower exposure. The input extracts were also subjected to immunoblotting (lower panels). Three independent experiments were done, and one representative experiment is shown.

B  CDKL5 phosphorylates CEP131 at Ser[35] in human cells. HEK293 cells were co-transfected with CDKL5 (wild type "WT" or kinase-dead K[42]R mutant) and/or FLAG-CEP131 (wild type WT or a S[35]A mutant). Anti-FLAG precipitates were subjected to Western blotting with the antibodies indicated. "Hi" higher exposure; "lo" lower exposure. The input extracts were also subjected to immunoblotting (lower panels). Three independent experiments were done, and one representative experiment is shown.

Source data are available online for this figure.

(pSer[159]), AFADIN (pSer[217]), FERMT2 (pSer[192]) and NUSAP1 (pThr[355])—do not (Fig 4A). This observation raised the possibility that the RPXSA motif might represent a consensus sequence for phosphorylation by CDKL5, and we next set out to investigate the importance of the determinants that influence serine phosphorylation by CDKL5. To this end, a range of synthetic peptides was synthesized, based on the sequence around MAP1S Ser[900]. The wild-type peptide sequence was KKRASRPLS[900]ARSEPSE (Fig 4C), and we tested the impact of making amino acid substitutions at Arg[897], Pro[898], Leu[899], Ser[900] and Ala[901]. As shown in Fig 4C, changing Arg[897] to the positively charged residue Lys or the negatively charged residues Asp or Glu causes a severe reduction in peptide

phosphorylation by CDKL5. Changing Pro[898] to Ala, Phe, Gly, Leu or the aromatic ring-bearing amino acids Tyr or His severely reduces Ser[900] phosphorylation, whereas substituting Leu[899], which is the variable "X" residue in the RPXSA motif, has no effect (Fig 4C). As expected, changing Ser[900] to Ala abolishes peptide phosphorylation although changing this residue to Thr permits peptide phosphorylation albeit at around 55% of the Ser-containing peptide. Changing Ala[901] to bulkier residues Val, Ile or Leu causes a severe reduction in Ser[900] phosphorylation, but changing Ala[901] to Gly, a small residue like Ala, permits Ser[900] phosphorylation at around 60% of wild-type levels (Fig 4C).

The data above indicate that the residues at Arg[897], Pro[898] and Ala[901] are important in allowing CDKL5 to phosphorylate Ser[900] of MAP1S, and they suggest that CDKL5 prefers to phosphorylate Ser residues in an RPX[S/T][A/G] motif. However, a report in the literature described robust phosphorylation of AMPH1 on Ser[293] by a recombinant form of the CDKL5 catalytic domain *in vitro* (Sekiguchi *et al*, 2013). AMPH1 Ser[293] lies in the sequence RPRS[293]P, suggesting that CDKL5 can accommodate a Pro residue C-terminal to the phosphorylated serine. We investigated this possibility in two ways. First, we found that CDKL5 can phosphorylate a synthetic peptide corresponding to the sequence around Ser[293] of AMPH1 (KKPAPVRPRS[293]PSQTRKG) with an efficiency similar to the MAP1S Ser[900] peptide; substituting Ser[293] for alanine abolished phosphorylation of this peptide (Fig 4D). Second, we found that substituting Ala[901] in the synthetic MAP1S Ser[900] peptide for a Pro residue allowed phosphorylation at around 80% of the wild-type peptide (Fig 4D). Taken together, these data suggest that the CDKL5 prefers substrates with serines that lie in the motif RPX[S/T][A/G/P] although it is possible that residues other than Ala/Gly/Pro can be accommodated C-terminal to the phospho-acceptor serine.

We next mined our original mass spectrometry data for phosphopeptides with phosphoserines in the motif RPX[S/T][A/G/P] that were more abundant in CDKL5-expressing cells than in knockout cells (Dataset EV1). In addition to MAP1S and CEP131, this analysis revealed phosphorylation of Ser[1115] of DLG5, a protein involved in signalling to the microtubule-based cytoskeleton (Liu *et al*, 2014; Kwan *et al*, 2016). To confirm that CDKL5 can phosphorylate DLG5 Ser[1115] in cells, FLAG-tagged human DLG5 was co-expressed with untagged CDKL5 (wild type or a K[42]R kinase-dead mutant) or empty vector in HEK293 cells. Extracts were subjected to immunoprecipitation with anti-FLAG antibodies, and phosphorylation site analysis was carried out by extracted ion chromatograms (XIC) analysis of the phosphopeptide containing Ser[1115]. FLAG-MAP1S and FLAG-CEP131 were analysed in parallel as positive controls. As shown in Fig 5A and B, FLAG-DLG5 was more highly phosphorylated on Ser[1115] when co-expressed with wild-type CDKL5, than with the kinase-dead equivalent or empty vector, and the change in DLG5 phosphorylation was comparable to that seen with MAP1S Ser[900] and CEP131 Ser[35] (Fig 5C–F). Furthermore, CDKL5 phosphorylated a synthetic peptide based on the sequence around DLG5 Ser[1115] even more robustly than the MAP1S Ser[900] peptide, and as expected, substituting Arg[1112], Pro[1113] or Ala[1116] severely reduced DLG5 peptide phosphorylation efficiency (Fig EV3). It is interesting to note that DLG5 Ser[1115] shows a high degree of evolutionary conservation among a range of orthologues, suggesting functional importance (Fig 5G). These data reveal DLG5 as a substrate of CDKL5.

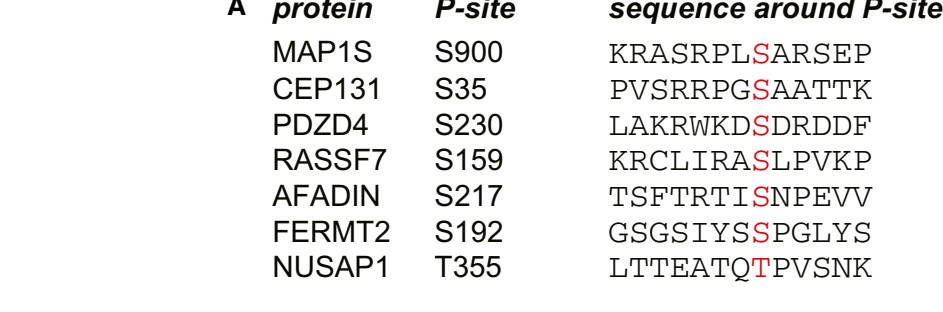

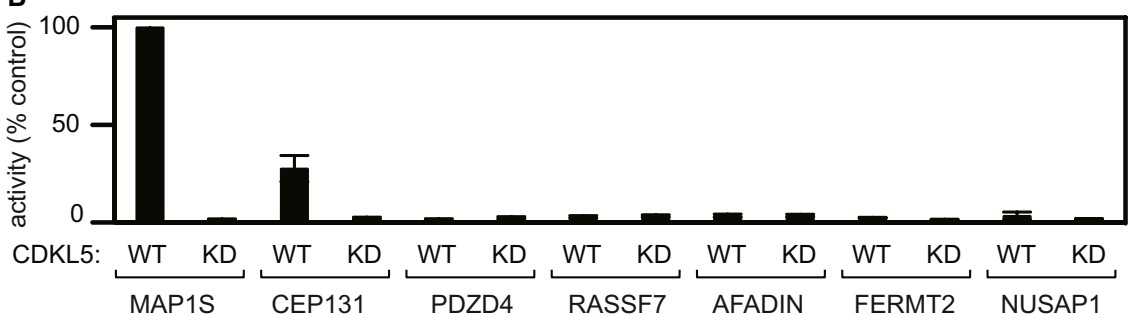

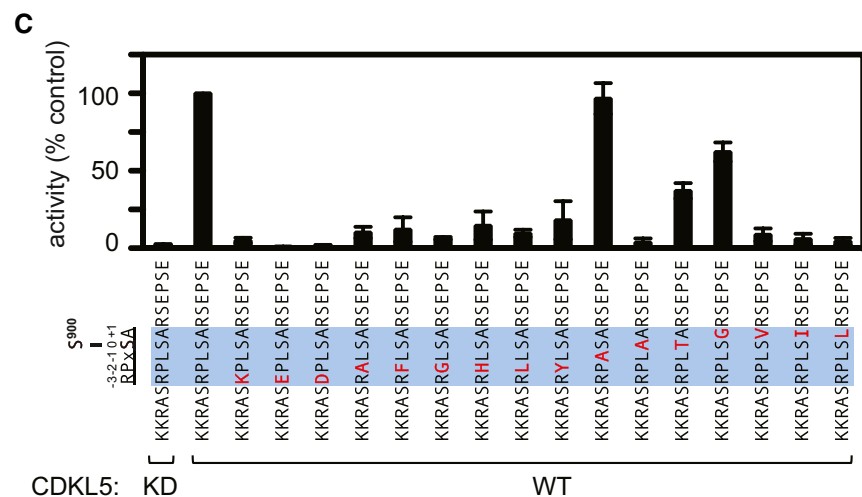

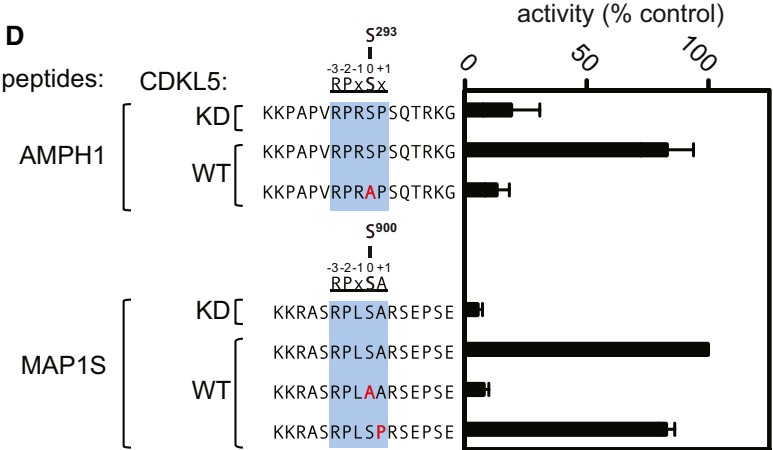

**Figure 4.**

## T-loop autophosphorylation controls CDKL5 activity in cells

T-loop phosphorylation is a key mechanism for kinase regulation, and CDKL5 has a Thr-Glu-Tyr (TEY) motif between catalytic subdomains VII and VIII that in MAPKs and CDKs must be phosphorylated for activity. Intriguingly, the $T^{169}$-E-$Y^{171}$ motif in CDKL5 is preceded by a conserved Tyr at residue 168 (Fig EV4). In experiments to map sites of CDKL5 phosphorylation, $Tyr^{171}$ emerged as a major site of tyrosine autophosphorylation (data not shown). Consistent with this idea, FLAG-CDKL5, but not the kinase-dead FLAG-CDKL5 $K^{42}R$ mutant, cross-reacted with anti-phospho-Tyr antibodies, before and after preincubation of precipitates with $Mg^{2+}$-ATP (Fig 6A), and this was prevented by mutating $Tyr^{171}$, but not $Tyr^{168}$ (Fig 6A). These data suggested that CDKL5 autophosphorylates on $Tyr^{171}$. To test this further, antibodies were raised in sheep against a phosphopeptide corresponding to the sequence around $Tyr^{171}$. As shown in Fig 6B, the affinity-purified antibodies cross-reacted with CDKL5 wild-type and a $Tyr^{168}Ala$ mutant, but not with a $Tyr^{171}Ala$ mutant. Furthermore, the kinase-inactivating $K^{42}R$ mutation also prevented CDKL5 $Tyr^{171}$ phosphorylation. These data strongly suggest that CDKL5 autophosphorylates on $Tyr^{171}$ in the T-loop. Surprisingly, we found that CDKL5 is incapable of phosphorylating $Tyr^{171}$-containing synthetic peptides based on the CDKL5 T-loop under conditions where MAP1S $Ser^{900}$ is phosphorylated efficiently (Fig 6C). Furthermore, substituting the serine phospho-acceptor residue in the MAP1S $Ser^{900}$ synthetic peptide for a tyrosine residue abolished peptide phosphorylation (Fig 6C). Taken together, these data indicate that although CDKL5 can catalyse intramolecular autophosphorylation on $Tyr^{171}$, it cannot phosphorylate exogenous substrates on tyrosine.

We next set out to test the effect of mutating $Tyr^{171}$—and also $Tyr^{168}$ and $Thr^{169}$ singly and in combination—on CDKL5 activity. To this end, HEK293 cells were co-transfected with FLAG–MAP1S (wild type "WT" or $Ser^{900}Ala$ "SA") and CDKL5 (wild type "WT", $K^{42}R$ kinase-dead "KD" mutant or T-loop mutants). FLAG precipitates were probed with MAP1S $pSer^{900}$ antibodies. As shown in Fig 6D, mutating $Tyr^{171}$ completely abolished CDKL5 activity towards MAP1S, whereas mutating $Tyr^{168}$ or $Thr^{169}$ to A had no apparent effect. The effect of T-loop mutations on intrinsic CDKL5 activity *in vitro* was also examined. As shown in Fig 6E, mutating $Tyr^{171}$ completely abolished CDKL5 activity towards the MAP1S $Ser^{900}$-containing peptide, whereas mutating $Tyr^{168}$ or $Thr^{169}$ to A had no effect. In a similar manner, mutating $Tyr^{171}$ prevented CDKL5-dependent phosphorylation of CEP131 on $Ser^{35}$ in cells and

abolished the activity of CDKL5 towards the CEP131 $Ser^{35}$-containing peptide *in vitro*, whereas mutating $Tyr^{168}$ or $Thr^{169}$ to A had no effect (Fig EV5A and B). Taken together, these data show that the activity of CDKL5 towards its substrates is critically dependent on autophosphorylation of $Tyr^{171}$ in the T-loop TEY motif.

## Pathogenic mutations reduce CDKL5 activity towards MAP1S and CEP131

The impact of pathogenic CDKL5 mutations on substrate phosphorylation in cells is unclear because the substrates have remained elusive. We next investigated the impact of pathogenic mutations on the activity of CDKL5 towards MAP1S. Most of the pathogenic mutations are located in the kinase catalytic domain (Fig 1A; Krishnaraj *et al*, 2017), and we investigated the impact of pathogenic kinase domain mutations $Gly^{20}Asp$ (Raymond *et al*, 2013), $Leu^{64}Pro$ (Fichou *et al*, 2009), $Ile^{72}Thr$ (Saletti *et al*, 2009), $Arg^{178}Trp$ (Nemos *et al*, 2009) and $Gln^{219}Pro$ (Hagebeuk *et al*, 2013). We also investigated a series of CDKL5 variants that are either benign: $Gln^{791}Pro$ (Tao *et al*, 2004) and $Val^{999}Met$ (Nectoux *et al*, 2006) or of uncertain significance: $Leu^{302}Phe$ (Liang *et al*, 2011), $Asn^{399}Thr$ (Sprovieri *et al*, 2009), $Val^{718}Met$ (Krishnaraj *et al*, 2017) and $Val^{793}Ala$ (Archer *et al*, 2006; Fig 1A). HEK293 cells were co-transfected with HA-tagged MAP1S and CDKL5 (wild type "WT", $K^{42}R$ kinase-dead "KD" or other mutants). Anti-HA precipitates were probed with MAP1S $pSer^{900}$ antibodies. As shown in Fig 7A, CDKL5 pathogenic kinase domain mutations $Gly^{20}Asp$, $Leu^{64}Pro$, $Ile^{72}Thr$, $Arg^{178}Trp$, and $Gln^{219}Pro$ caused a severe reduction in MAP1S $Ser^{900}$ phosphorylation. In a similar vein, these mutations caused a major reduction in the activity of FLAG-CDKL5 precipitates towards the synthetic MAP1S $Ser^{900}$ peptide (Fig 7B). The $Leu^{302}Phe$, $Asn^{399}Thr$, $Val^{718}Met$, $Gln^{791}Pro$, $Val^{793}Ala$ and $Val^{999}Met$ variants that are either benign or of uncertain significance to CDKL5 disorder supported wild-type levels of MAP1S phosphorylation (Fig 7A). Furthermore, these amino acid substitutions did not affect the activity of CDKL5 towards the synthetic MAP1S $Ser^{900}$ peptide (Fig 7B).

We also investigated the impact of pathogenic mutations on the activity of CDKL5 towards CEP131. HEK293 cells were co-transfected with FLAG-tagged CEP131 and CDKL5 (wild type "WT", $K^{42}R$ kinase-dead "KD" or other mutants). Anti-FLAG precipitates were probed with CEP131 $pSer^{35}$ antibodies. As shown in Fig 7C, CDKL5 pathogenic mutations $Gly^{20}Asp$, $Leu^{64}Pro$, $Ile^{72}Thr$ and $Gln^{219}Pro$ caused a severe reduction in

---

**Figure 4. CDKL5 sequence determinants favouring phosphorylation by CDKL5.**

A  Sequence of synthetic peptides surrounding the sites of phosphorylation in putative CDKL5 substrates that were identified in the phosphoproteomic screening. The amino acid number of the phosphorylated residue in each peptide (highlighted in red) is listed.

B  Peptide kinase assays to investigate CDKL5 sequence specificity. Anti-FLAG precipitates from HEK293 cells transiently expressing FLAG-tagged CDKL5 (wild type "WT" or a $K^{42}R$ kinase-dead "KD" mutant) were incubated with the synthetic peptides from the proteins indicated (sequences shown in A) in the presence of $[\gamma\text{-}^{32}P]$-labelled ATP-$Mg^{2+}$, and peptide phosphorylation was measured by Cerenkov counting.

C  Same as (B), except that the peptides used were designed specifically to investigate the effect of amino acid substitutions at $R^{897}$, $P^{898}$, $L^{899}$ and $A^{901}$ on the phosphorylation of MAP1S $Ser^{900}$. The RPXSA motif is shaded in blue, and amino acid substitutions compared with the wild-type MAP1S $Ser^{900}$ peptide are shown in red.

D  Same as (C), except that phosphorylation of the indicated peptides from AMPH1 and MAP1S was compared.

Data information: In (B–D), phosphorylation of the control wild-type MAP1S peptide is taken as 100%. The data are represented as mean ± SEM from three independent experiments.

Source data are available online for this figure.

CEP131 Ser[35] phosphorylation. In a similar vein, these mutations caused a major reduction in the activity of FLAG-CDKL5 precipitates towards the synthetic CEP131 Ser[35] peptide (Fig 7D). In contrast, the Leu[302]Phe, Asn[399]Thr, Val[718]Met, Gln[791]Pro, Val[793]Ala and Val[999]Met variants that are either benign or of uncertain significance to CDKL5 disorder supported wild-type levels of CEP131 peptide phosphorylation (Fig 7C). Furthermore, these amino acid substitutions did not affect the activity of CDKL5 towards the synthetic CEP131 Ser[900] peptide (Fig 7D). Taken together, these data show that CDKL5 mutations causative of CDKL5 disorder cause a major reduction in CDKL5 activity towards MAP1S and CEP131. It is interesting to note that mutations Gly[20]Asp and Leu[64]Pro cause a severe reduction in CDKL5 autophosphorylation, but the Arg[178]Trp mutation that abolishes CDKL5 activity towards MAP1S and CEP131 has no apparent effect on CDKL5 autophosphorylation (Fig EV6). It is possible that Arg[178] is involved in exogenous substrate binding, which could account for the discrepancy.

## Discussion

In this study, we used state-of-the-art phosphoproteomic screening to identify, and then validated, the first cellular substrates of CDKL5: MAP1S, CEP131 and DLG5. The phosphorylated serine in all three substrates lies in the motif RPXSA, although experiments with synthetic peptides revealed that substitution of S for T allowed MAP1S peptide phosphorylation at around 55% of the S-containing peptide (Fig 4C), and substitution of A for G or P allowed peptide phosphorylation at around 60 and 90% of the wild-type sequence, respectively. Therefore, CDKL5 can phosphorylate Ser residues that lie in the motif RPX[S/T][A/G/P], although it is possible that residues other than A/G/P can be tolerated C-terminal to the phospho-acceptor serine.

It is interesting to note that in phospho-motif analysis SP motifs in general were highly enriched in the phosphopeptides that were more abundant in CDKL5-expressing cells (Appendix Fig S3B). This might not be surprising given that CDKL5 shows a high degree of similarity in the catalytic domain with MAPKs and CDKs which phosphorylate target proteins on [S/T]P motifs. However, it is unlikely that CDKL5 catalyses direct phosphorylation of targets on SP motifs unless they lie in an RPXSP motif. Under conditions where CDKL5 can

phosphorylate peptides containing MAP1S Ser[900] or CEP131 Ser[35], it cannot phosphorylate peptides containing FERMT2 Ser[192] or NUSAP1 Thr[355] which contain an SP and TP motif, respectively (Fig 4A and B). Moreover, AMPH1 Ser[293] lies in an RPRSP motif and substituting Arg[290] for Lys in the AMPH1 Ser[293]-containing synthetic peptide abolishes phosphorylation by CDKL5 (data not shown). Nonetheless, phospho-SP motifs are more abundant in CDKL5-expressing cells than in CDKL5 knockout cells (Dataset EV1), and it may be that CDKL5 influences the activity of one or more proline-directed kinase, directly or indirectly. It is interesting to note that all of the phosphorylation sites identified in our phosphoproteomic analysis in CDKL5 itself are SP motifs (Fig 2A; Dataset EV1). Preliminary analysis indicates that these are not sites of autophosphorylation, however, and are instead catalysed by other kinases (data not shown). On the other hand, CDKL5 appears to autophosphorylate on Tyr[171] even though it cannot phosphorylate exogenous substrates on tyrosine. CDKL5 autophosphorylation appears to be critical for kinase activity. It will be interesting to investigate the regulation of CDKL5 T-loop phosphorylation in cells and to explore the underlying mechanisms.

The substrates of CDKL5 we validated are all involved in control of the cytoskeleton, suggesting that CDKL5 may regulate cytoskeletal function. MAP1S is a ubiquitously expressed protein that can bind to and stabilize microtubules *in vitro* and in cells (Orban-Nemeth *et al*, 2005; Ding *et al*, 2006; Dallol *et al*, 2007; Tegha-Dunghu *et al*, 2014; Mohan & John, 2015), and depletion of MAP1S causes mitotic abnormalities (Dallol *et al*, 2007). It has been reported that MAP1S bridges microtubules with mitochondria and components of the autophagy machinery, and consequently, cells from *Map1s*[−/−] mice show decreased levels of autophagosomal biogenesis and clearance of damaged mitochondria (Xie *et al*, 2011a,b). The N-methyl-*D*-aspartate (NMDA) receptor subunit NR3A also interacts with MAP1S in brain, and thus, MAP1S might link NR3A to the microtubule cytoskeleton (Eriksson *et al*, 2007). In this light, the NMDA receptor-mediated synaptic response is enhanced in the hippocampus of *Cdkl5* knockout mice, and the N2B subunit of the NMDA receptor overaccumulates in post-synaptic densities of hippocampal neurons in these mice (Okuda *et al*, 2017). Furthermore, the enhanced seizure susceptibility seen in *Cdkl5* knockout mice is abrogated by an NMDA receptor antagonist (Okuda *et al*, 2017). Intriguingly, Ser[900] in MAP1S lies close to one of two reported

---

**Figure 5.    Mass spectrometric validation of DLG5 as a CDKL5 substrate.**

A–F    Testing CDKL5-dependent DLG5 phosphorylation in cells. HEK293 cells were co-transfected with FLAG-DLG5 (A, B), FLAG-MAP1S (C, D) or FLAG-CEP131 (E, F) together with either empty vector (−), wild-type CDKL5 (WT) or kinase-dead CDKL5 (K[42]R). Cells were harvested after 24 h and lysed, and extracts were subjected to immunoprecipitation with anti-FLAG-agarose beads. Precipitates were resolved by SDS–PAGE and stained with Coomassie Brilliant Blue (COO, top panel). A small fraction of the precipitates was subjected to Western blotting with anti-FLAG antibodies (A, C). The input extracts were also subjected to immunoblotting with the antibodies indicated in the lower panels. Three independent experiments were done, and one representative experiment is shown. Coomassie-stained bands as shown in top panels of (A, C and E) were excised, destained and proteins digested. Peptides were extracted, TMT labelled and analysed using mass spectrometry. VSN-calibrated and transformed, isotopically corrected reporter ion intensities of the phosphopeptides of interest are plotted in (B, D and F); the outline of the box plot indicates minimum and maximum values, and the middle line indicates the median. Higher intensity corresponds to higher phosphopeptide abundance in the relevant sample. Statistical testing was carried out using a *t*-test (Computer Code EV1; Appendix Fig S4); to account for multiple testing, the significance threshold was adjusted from α = 0.05 to α = 0.00833 (six *t*-tests) by Bonferroni correction (Computer Code EV1). All *t*-tests resulted in a *P*-value below the adjusted significance threshold. \*\**P* < 0.01; \*\*\**P* < 0.001. Individual *P*-values are shown in Table EV3.

G    Sequence alignment of the Ser[1115] phosphorylation site in DLG5 in the species indicated.

Source data are available online for this figure.

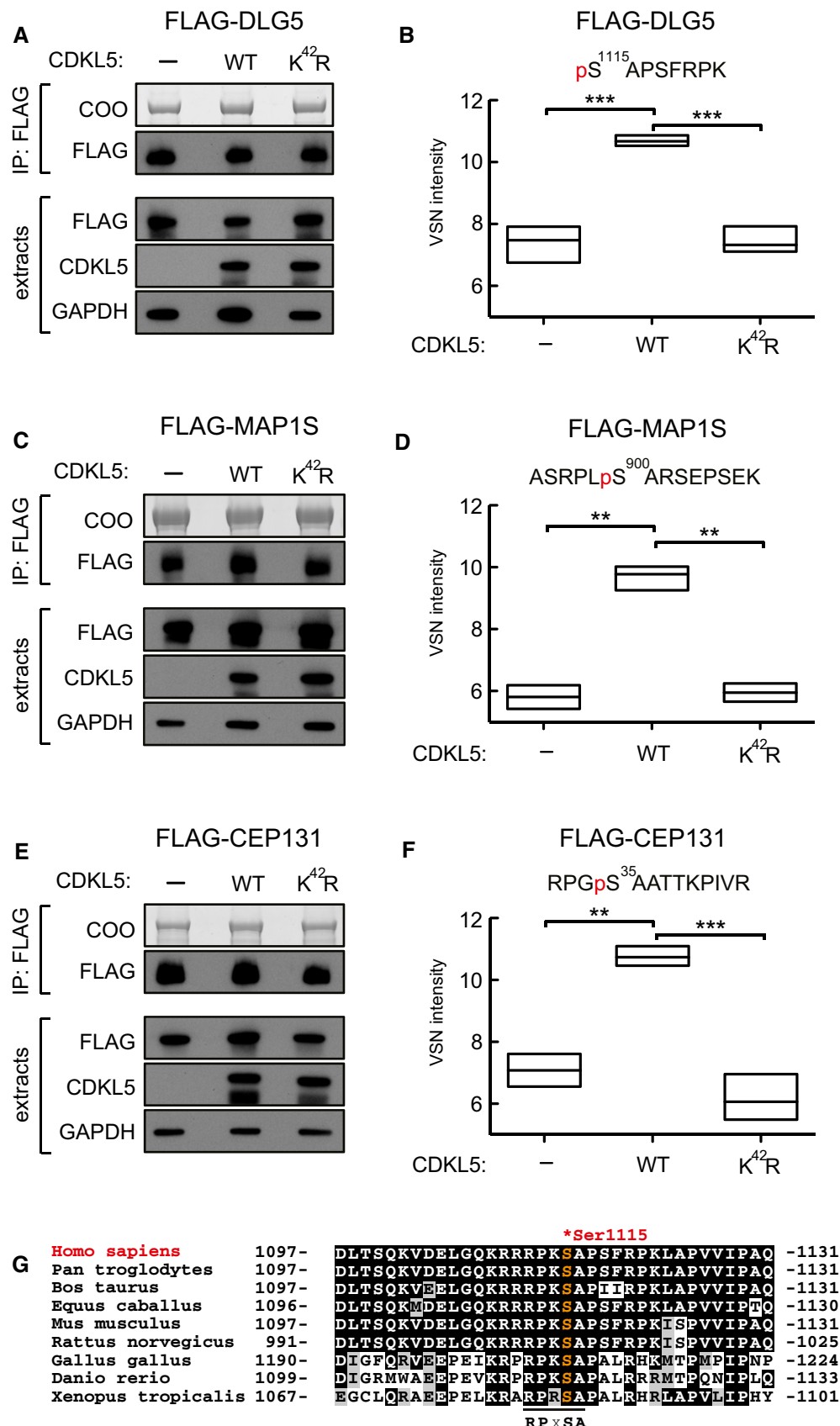

**Figure 5.**

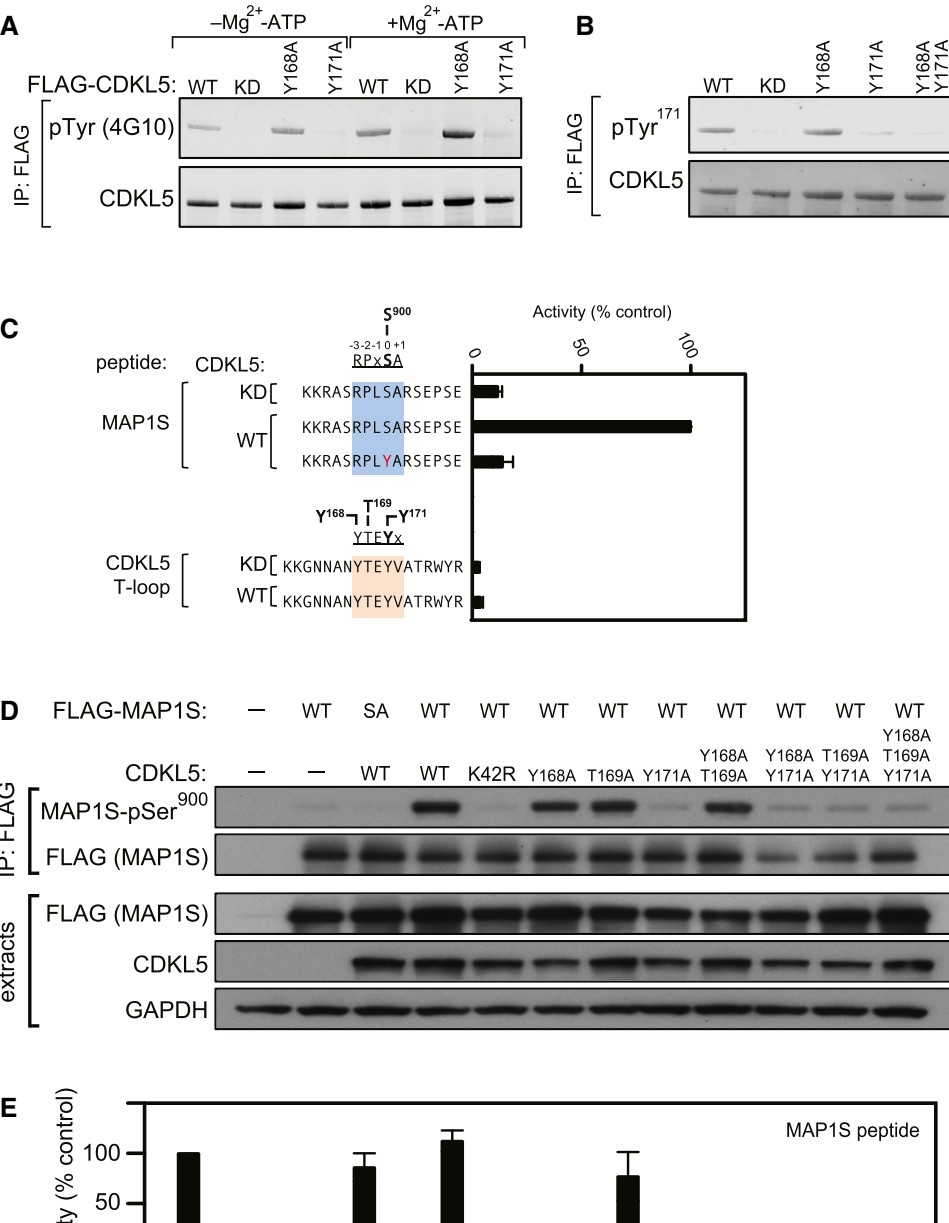

**Figure 6.  T-loop autophosphorylation is critical for CDKL5 activity.**

A, B   Tyr-autophosphorylation of CDKL5. Anti-FLAG precipitates from HEK293 cells transiently expressing FLAG-tagged CDKL5 (wild type "WT" or mutants K[42]R kinase-dead "KD", Y[168]A,Y[171]A or both) were subjected to immunoblotting with the antibodies indicated before or after incubation of precipitates with Mg[2+]-ATP for 30 min at 30°C. Each experiment was done three times, and a representative experiment is shown.

C   CDKL5 cannot phosphorylate Tyr-containing synthetic peptides. Anti-FLAG precipitates from HEK293 cells transiently expressing FLAG-tagged CDKL5 (wild type "WT" or a K[42]R kinase-dead "KD" mutant) were incubated with the synthetic peptides indicated in the presence of [γ-[32]P]-labelled ATP-Mg[2+], and peptide phosphorylation was measured by Cerenkov counting. Data are represented as mean ± SEM from three independent experiments.

D   HEK293 cells were co-transfected with untagged CDKL5 (wild type "WT" or the mutants indicated) and FLAG-tagged MAP1S [wild type (WT), a Ser[900]Ala mutant (SA) or empty vector (−)]. Anti-FLAG precipitates were subjected to Western blotting with the antibodies indicated. The input extracts were also subjected to immunoblotting (lower panels). Each experiment was done three times, and a representative example is shown.

E   Anti-FLAG precipitates from HEK293 cells transiently expressing FLAG-tagged CDKL5 (wild type "WT" or the mutants indicated) were incubated with a synthetic peptide corresponding to the sequence around the MAP1S Ser[900] phosphorylation site, in the presence of [γ-[32]P]-labelled ATP-Mg[2+]. Peptide phosphorylation was quantitated in a scintillation counter. Data are represented as mean ± SEM from three independent experiments.

Source data are available online for this figure.

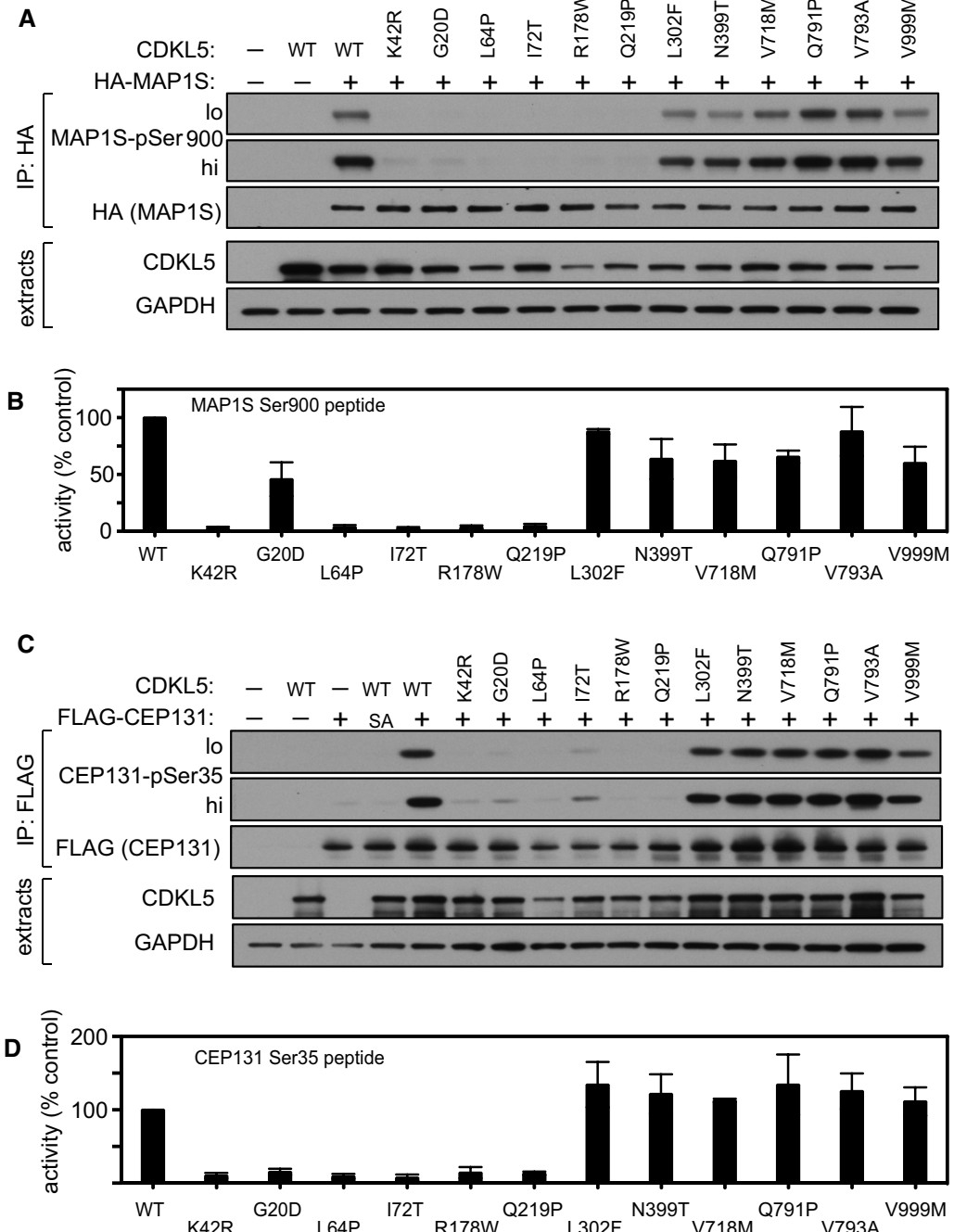

**Figure 7. Effect of pathogenic CDKL5 mutations on CDKL5 activity.**

A   HEK293 cells were co-transfected with untagged CDKL5 (wild type "WT" or the mutants indicated) and HA-tagged MAP1S. Anti-HA precipitates were subjected to Western blotting with the antibodies indicated. The input extracts were also subjected to immunoblotting (lower panels). Three independent experiments were done, and one representative experiment is shown. Hi, high exposure; lo, low exposure.

B   Anti-FLAG precipitates from HEK293 cells transiently expressing FLAG-tagged CDKL5 (wild type "WT" or the mutants indicated) were incubated with a synthetic peptide corresponding to the sequence around the MAP1S Ser$^{900}$ phosphorylation site, in the presence of [γ-$^{32}$P]-labelled ATP-Mg$^{2+}$. Peptide phosphorylation was quantitated in a scintillation counter. Data are represented as mean ± SEM from three independent experiments.

C   HEK293 cells were co-transfected with untagged CDKL5 (wild type "WT" or the mutants indicated) and FLAG-tagged CEP131. Anti-FLAG precipitates were subjected to Western blotting with the antibodies indicated. The input extracts were also subjected to immunoblotting (lower panels). Three independent experiments were done, and one representative experiment is shown. Hi, high exposure; lo, low exposure.

D   Anti-FLAG precipitates from HEK293 cells transiently expressing FLAG-tagged CDKL5 (wild type "WT" or the mutants indicated) were incubated with a synthetic peptide corresponding to the sequence around the CEP131 Ser$^{35}$ phosphorylation site, in the presence of [γ-$^{32}$P]-labelled ATP-Mg$^{2+}$. Peptide phosphorylation was quantitated in a scintillation counter. Data are represented as mean ± SEM from three independent experiments.

Source data are available online for this figure.

microtubule-binding domains (Orban-Nemeth *et al*, 2005; Ding *et al*, 2006; Dallol *et al*, 2007). From this point of view, it is possible that MAP1S Ser[900] phosphorylation by CDKL5 alters the microtubule-binding capacity of MAP1S which could in turn impact on the trafficking of mitochondria, or neurotransmitter-containing vesicles or the NMDA receptor, for example, in a way required to prevent neuronal deterioration in CDKL5 disorder.

CEP131 is a centrosomal protein which has been implicated in the formation and function of primary cilia. In this light, depletion of CEP131 from human epithelial cells causes reduced ciliogenesis and defects in cilial morphogenesis (Graser *et al*, 2007). CEP131 depletion was also reported to cause reduced proliferation rate, centriole amplification, chromosomal instability and DNA damage (Staples *et al*, 2012). Intriguingly, CEP131 was reported to be phosphorylated by the stress-inducible MK2 kinase in response to UV light on both Ser[47] and Ser[78]. Phosphorylation of CEP131 on these residues leads to 14-3-3 binding, sequestration of CEP131 in the cytosol away from centrosomes, and abolishes stress-induced centriolar satellite reorganization (Villumsen *et al*, 2013; Tollenaere *et al*, 2015). Ser[35] targeted by CDKL5 lies close to the MK2 phosphorylation sites and may therefore impact on stress-induced centriolar satellite reorganization or other aspects of centriolar and cilial function. It is interesting to note that *Drosophila* and zebrafish lacking the relevant CEP131 orthologue show phenotypes reminiscent of human ciliopathies (Andersen *et al*, 2003; Wilkinson *et al*, 2009; Ma & Jarman, 2011). Ciliopathies in humans are most commonly associated with retinal and renal problems, which are not associated with CDKL5 disorder, and therefore, it is unlikely that CDKL5 disorder is a ciliopathy. However, a subset of ciliopathies affect the brain and cause seizures—juvenile myoclonic epilepsy, for example. Intriguing connections between CDKL5 and primary cilia have been reported. CDKL5 (LF5p) controls ciliary length in *Chlamydomonas* (Tam *et al*, 2013), and human CDKL5 localizes to cilia and impairs ciliogenesis when overexpressed (Canning *et al*, 2018). It will be important to test cilium function in CDKL5 knockout cells and in the context of pathogenic mutations, and to check the impact of CDKL5-mediated phosphorylation of CEP131.

There are several interesting applications made possible by the identification of the first cellular substrates of CDKL5. Firstly, phospho-MAP1S and phospho-CEP131 antibodies can be used as biomarkers of CDKL5 activity. This will enable experiments to screen for stimuli that activate CDKL5, for example, and will allow the effectiveness of new treatments for CDKL5 disorder to be tested—gene replacement therapy and drugs that promote reading through nonsense mutations, for example. A major limitation, however, is that better antibodies will be required to allow analysis of the phosphorylation of endogenous MAP1S and CEP131. Recent identification of the Rab proteins as targets of the Parkinson's disease kinase LRRK2 required the development of rabbit monoclonal antibodies to allow analysis of endogenous Rab phosphorylation in biological samples (Steger *et al*, 2017; Lis *et al*, 2018). Our data demonstrate for the first time that pathogenic mutations in CDKL5 are loss of function in that they inhibit phosphorylation of MAP1S and CEP131. Highly sensitive phospho-specific antibodies may allow screening for small molecules, or genes that when deleted by genome editing rescue the phosphorylation of MAP1S and CEP131 phosphorylation in CDKL5-defective cells. Such an endeavour may pave the way for new therapies to treat CDKL5 disorder.

# Materials and Methods

All DNA constructs and antibodies generated for the present study, and datasheets for each plasmid, can be requested via our reagents website (http://mrcppureagents.dundee.ac.uk/reagents-from-paper/rouse-paper-1). A list of plasmids, oligonucleotides and peptides used in this study is shown in Table EV1.

### Reagents

Iodoacetamide (I1149), microcystin-LR solution (33893) and Phosphatase Inhibitor Cocktail-2 (P5726) solution were all purchased from Sigma. Universal Nuclease for Cell Lysis (88702) was purchased from Thermo Fisher Scientific and EDTA-free Protease Inhibitor Cocktail (11873580001) from Roche. All peptides were purchased from Peptides & Elephants.

### Antibodies

The following antibodies were raised by the Division of Signal Transduction Therapy at the University of Dundee in sheep and purified against the relevant antigens: anti-CDKL5 (S957D; raised against amino acids 350–650 of human CDKL5 expressed in bacteria using plasmid DU50406; third bleed); anti-MAP1S phospho-Ser[900] (SA339; raised against the peptide KKRASRPLpS[900]ARSEPSE conjugated to bovine serum albumin; third bleed); anti-CEP131 phospho-Ser[35] (SA373; raised against the peptide CKKPPVSRRPGpS[35]AATTKP conjugated to bovine serum albumin; third bleed); anti-CDKL5 phospho-Tyr[171] (SA547; raised against the peptide GNNANYTEpY[171]VATRWYR conjugated to bovine serum albumin; third bleed). Sheep were immunised with each antigen followed by up to five further injections 28 days apart, with bleeds performed 7 days after each injection.

Anti-FLAG (M2) and anti-HA antibodies were obtained from Sigma, and anti-GAPDH (14C10) antibodies were purchased from Cell Signalling. Secondary anti-sheep, anti-mouse and anti-rabbit HRP-conjugated antibodies were purchased from Life Technologies and were used at 1:5,000–1:10,000 dilutions, for 1 h at room temperature. Anti-pTyr antibodies (4G10) were from Merck.

### Cell culture

All cells were kept at 37°C under humidified conditions with 5% $CO_2$. HEK293, HEK293T and U2OS Flp-In T-Rex (FRT) cells were grown in DMEM supplemented with 10% (v/v) foetal bovine serum, 100 U/ml penicillin, 100 μg/ml streptomycin, 1% (v/v) L-glutamate (GIBCO, Invitrogen), 1% (v/v) sodium pyruvate and 1% (v/v) non-essential amino acids. U2OS FRT cells were maintained in 10 μg/ml blasticidin. Hygromycin (100 μg/ml) was used to select for the integration of constructs in Flp-In recombination sites.

### Cell lysis, immunoprecipitation, SDS–PAGE and Western blotting

Cells were lysed in a 50 mM Tris/HCl (pH 7.4) buffer containing 0.27 M sucrose, 150 mM NaCl, 1% (v/v) Triton X-100, 0.5% (v/v) Nonidet NP-40 and 0.1% (v/v) 2-mercaptoethanol. Lysis buffer was supplemented with a protease inhibitor cocktail (cOmplete™, EDTA-free Protease Inhibitor Cocktail), benzonase (Novagen, 50 U/ml),

microcystin-LR (Cat. Number, 33893, Sigma) at a final concentration of 10 ng/ml and phosphatase inhibitor cocktail-2 (P5726, Merck) at 1% (v/v). Protein extracts were then incubated for 30 min at 4°C and clarified by centrifugation at 17,000 *g* in a refrigerated tabletop centrifuge. To immunoprecipitate endogenous CDKL5, sheep polyclonal antibodies were incubated overnight with protein G-agarose (from Expedeon; number APG 0100) at 1 μg antibody per 10 μl of settled beads, and then washed twice in PBS. Approximately 100–500 μg of whole-cell extracts were incubated with 10 μl (settled) of CDKL5-containing beads for 2–3 h at 4°C. For anti-FLAG or anti-HA immunoprecipitations, approximately 10 μl (settled volume) of the following resins was used per 1 mg extract: FLAG-M2 agarose (Sigma-Aldrich; F1804) or HA-agarose (Sigma-Aldrich; A2095) beads for 2–3 h at 4°C. Precipitates were washed three times with lysis buffer and once in cold PBS before boiling at 95°C in LDS–PAGE sample buffer (Thermo Fisher) containing 5% (v/v) 2-mercaptoethanol. Samples were resolved in 4–12% Bis–Tris SDS–PAGE gradient gels (NuPAGE, Thermo Fisher).

For Western blotting with phospho-specific MAP1S pSer[900] and CEP131 pSer[35] antibodies, the antibodies were diluted in 5% (w/v) non-fat dry milk in TBS-Tween (0.2% v/v) at a final concentration of 0.2 μg/ml and supplemented with 10 μg/ml of the corresponding non-phospho peptide; this mixture was incubated for 1 h at room temperature. Membranes were incubated overnight at 4°C with the relevant antibody:non-phosphopeptide mixture. Rabbit anti-sheep HRP-conjugated secondary antibodies were used at a 1:5,000 dilution, for 1 h at room temperature.

## Transfections

Unless otherwise stated, HEK293 cells were transfected using the calcium phosphate transfection protocol. HEK293 cells were seeded at 20–30% confluency into 15-cm plates and transfected 24 h later. Briefly, for each transfection around 1 ml of 2× HEPES-buffered saline (HBS) solution was mixed dropwise with 1 ml of a solution containing 122 μl 2 M calcium chloride and the appropriate amount of plasmid DNA: 10 μg of DNA when only one plasmid was used, or 5 μg each when cells were co-transfected with two different plasmids. The transfection mixture was added dropwise on top of cells, and plates were incubated for another 24 h before being harvested in cold PBS buffer and lysed immediately after (as stated below). To prepare 2× HBS solution, sodium phosphate dibasic heptahydrate from Sigma-Aldrich (cat. Number 431478) was used.

## CRISPR-mediated disruption of CDKL5 in U2OS Flp-In T-REx cells

In order to knock out CDKL5 from cells, we used a modified Cas9 D[10]A double nickase system (Ran *et al*, 2013) that requires the presence of two appropriately offset single-guide RNA pairs. We identified two sgRNA sequences in exon 5 of human CDKL5 gene separated by nine nucleotides, as shown in Appendix Fig S1. Single-guide RNA CDKL5 sense and single-guide RNA CDKL5 antisense sequences were cloned in their respective plasmids to generate pBABED-PURO-U6-CDKL5-sense and pX335-CAS9 D10A-CDKL5-antisense. 1 μg of each plasmid was used to transfect U2OS FRT cells. 24 h post-transfection, medium was replaced with fresh medium containing 2 μg/ml puromycin and cells were allowed to grow for another 48 h in selection conditions. Approximately 3,000 cells were then seeded into 15 cm

plates and allowed to grow until they generated visible colonies, and 60 viable individual colonies were picked and expanded. To screen for the loss of CDKL5 expression in 35 of these clones, cultures were lysed and CDKL5 was immunoprecipitated using sheep polyclonal CDKL5 antibodies (S957D). Precipitates were then immunoblotted, and loss of CDKL5 was confirmed using both a sheep polyclonal CDKL5 antibody and a mouse monoclonal antibody (D-12, Santa Cruz Biotechnologies). Five individual clones that showed no detectable CDKL5 protein in immunoprecipitations and Western blotting were selected, and genomic DNA around exon 5 was amplified by PCR using plasmids CDKL5-KO-FWD and CDKL5-KO-REV, cloned using the Stratalone PCR cloning kit (Stratagene) and sequenced using T7 and T3 oligonucleotides. Two clones (CDKL5Δ-7 and CDKL5Δ-13) lacked a wild-type *CDKL5* allele but instead harboured frameshift mutations in leading to missense and/or premature stop codons (Appendix Fig S1).

## Generation of stable cell lines using the Flp-In T-Rex system

To generate U2OS Flp-In T-REx cells stably expressing CDKL5, cells were co-transfected with 9 μg of POG44 Flp-recombinase expression vector (Thermo Fisher) and 1 μg of pcDNA5 FRT/TO-CDKL5, using GeneJuice Transfection Reagent (Millipore). Around 48 h after transfection, cells were selected in the presence of 100 μg/ml hygromycin and 10 μg/ml blasticidin in the medium. Ten to 12 days later, surviving colonies were pooled together and resulting cultures were analysed for the expression of CDKL5 after induction with increasing amounts of tetracycline hydrochloride (T3383, Sigma-Aldrich).

## Phosphoproteomic analysis

The global phosphoproteomics mass spectrometry data relating to Figs 1 and 2 have been deposited to the ProteomeXchange Consortium via the PRIDE partner repository (Jarnuczak & Vizcaino, 2017) with the dataset identifier PXD009374.

### Protein extraction and digestion for mass spectrometry
Three biological replicates for both CDKL5 knockout U2OS cells (clone 13) and the same cells in which CDKL5 was reintroduced were lysed in 8 M urea and 50 mM triethylammonium bicarbonate (TEAB; pH 8.5) and sonicated, and protein quantification was determined using the BCA Protein Assay Kit (Pierce Protein). Approximately 5 mg of protein extract for each sample was reduced by adding 5 mM Tris(2-carboxyethyl)phosphine (TCEP) for 30 min at room temperature, and cysteines were alkylated by adding 20 mM iodoacetamide for 30 min in the dark. Samples were then diluted to 1 M urea and digested overnight at 37°C by adding porcine trypsin (1:50, w/w; Pierce). Peptides were desalted and concentrated via C18 SPE on Sep-Pak cartridges (Waters).

### Phosphopeptide enrichment for global phosphoproteomic analyses
Phosphopeptide enrichment was similar to previously described (Larsen *et al*, 2005; Trost *et al*, 2009, 2012; Lai *et al*, 2015). Tryptic peptides (5 mg per TMT channel) were resuspended in 1 ml of 2 M lactic acid/50% (v/v) acetonitrile and centrifuged 15,000 × *g* for 20 min. Supernatants were placed in an Eppendorf LoBind Tube containing 24 mg of titanium dioxide beads (GL Sciences) and vortexed for 1 h at room temperature. Beads were washed twice

with 2 M lactic acid/50% (v/v) acetonitrile and once with 0.1% (v/v) TFA in 50% (v/v) acetonitrile. Phosphopeptides were eluted twice with 150 μl of 60 mM ammonium hydroxide (pH > 10.5), then acidified with 40 μl of 20% (v/v) formic acid and desalted via C18 Macro SpinColumns (Harvard Apparatus).

### Labelling tryptic peptides with tandem mass tags

Isobaric labelling of phosphorylated peptides was performed using the 6-plex tandem mass tag (TMT) reagents (Thermo Fisher). TMT reagents (0.8 mg) were dissolved in 41 μl of acetonitrile, and 20 μl was added to the corresponding fractions, previously dissolved in 50 μl of 50 mM TEAB, pH 8.5. The reaction was quenched by addition of 4 μl of 5% hydroxylamine after 1-h incubation at room temperature. Labelled peptides were combined, acidified with 200 μl of 1% TFA (pH ~ 2) and concentrated using C18 SPE on Sep-Pak cartridges (Waters). Labelling efficiency was tested for each TMT-labelled sample and was > 95%.

### High-pH reversed-phase chromatography fractionation

TMT-labelled phosphorylated peptides were combined at equal amounts for each channel and subjected to HPRP. Labelled peptides were solubilized in 20 mM ammonium formate (pH 8.0) and separated on a Gemini C18 column (250 × 3 mm, 3 μm C18 110 Å pore size; Phenomenex). Using a DGP-3600BM pump system equipped with a SRD-3600 degasser (Thermo Fisher), a 40 min gradient from 1 to 90% acetonitrile (flow rate of 0.25 ml/min) separated the peptide mixtures into a total of 40 fractions. The 40 fractions were merged into 12 samples, acidified with 1% (v/v) TFA (pH ~ 2), desalted via C18 Macro SpinColumns (Harvard Apparatus), dried under vacuum centrifugation and resuspended in 2% (v/v) ACN/0.1% (v/v) TFA for LC-MS/MS analysis.

### Liquid chromatography and tandem mass spectrometry (LC-MS/MS)

Fractions were separated on an Ultimate 3000 Rapid Separation LC Systems chromatography (Thermo Fisher) with a C18 PepMap, serving as a trapping column (2 cm × 100 μm ID, PepMap C18, 5 μm particles, 100 Å pore size) followed by a 50-cm EASY-Spray column (50 cm × 75 μm ID, PepMap C18, 2 μm particles, 100 Å pore size; Thermo Fisher) with a linear gradient of 2.4–20% (v/v; ACN, 0.1% (v/v) formic acid (FA)) over 175 min followed by a step from 20 to 28% (v/v) ACN, 0.1% (v/v) FA over 30 min, both segments had a flow rate at 300 nl/min. Mass spectrometric analysis was performed on an Orbitrap Fusion Tribrid mass spectrometer (Thermo Fisher) operated in "Top Speed" data-dependent, positive ion mode. FullScan spectra were acquired in a range from 400 to 1,600 $m/z$, at a resolution of 120,000 (at 200 $m/z$), with an automated gain control (AGC) of 300,000 ions and a maximum injection time of 50 ms. Precursors with a charge state of 1 were excluded. Intensity threshold for MS/MS fragmentation was set to $10^4$ counts. The most intense precursor ions were isolated with a quadrupole mass filter width of 1.6 $m/z$, and HCD fragmentation was performed with a one-step collision energy of 37.5% and activation Q of 0.25. MS/MS fragment ions were measured in the Orbitrap mass analyser with a 50,000 resolution at 200 $m/z$. The detection of MS/MS fragments was set up as the "Universal Method", using a maximum injection time of 300 ms and a maximum AGC of 2,000 ions.

### Data processing and quantitative data analysis of TMT data

Protein identification and TMT quantification were performed using MaxQuant version 1.5.1.7 (Cox & Mann, 2008). Trypsin/P was set as protease; stable modification: carbamidomethyl (C); variable modifications: oxidation (M), acetyl (protein N-terminal), phospho (STY); maximum eight modifications per peptide and two missed cleavages. Searches were conducted using a combined UniProt-TrEMBL Homo sapiens database with isoforms downloaded on July 15, 2015, plus common contaminants (42,095 sequences). Identifications were filtered at a 1% FDR at the peptide level, accepting a minimum peptide length of 5. TMT intensities were extracted, normalized for each condition and were used for downstream analyses in Perseus 1.5.3.1 (Cox & Mann, 2012). Student's *t*-test (two-tailed, homoscedastic) was performed on the normalized TMT intensities, and phosphopeptides with $P < 0.05$ and a fold change > 1.5 were considered significantly altered in abundance between the samples.

### Motif-X and network analysis

Significant mass spectrometry hits (> 1.5-fold change, $P < 0.05$) were analysed using UniProt, motif-x and the STRING database. Motif-x software (Schwartz & Gygi, 2005) was used with pre-aligned sequence tags around the phosphosites. Following parameters were applied: occurrences 10; significance $P < 0.0001$. String database v10.0 (Szklarczyk *et al*, 2015) analysis was performed using default parameters (medium confidence). The networks were exported into Cytoscape (Shannon *et al*, 2003).

## Extracted ion chromatogram (XIC) analysis

The mass spectrometry proteomics data have been deposited to the ProteomeXchange Consortium via the PRIDE partner repository (Jarnuczak & Vizcaino, 2017) with the dataset identifier PXD009327.

### In-gel digest

Bands from SDS–PAGE gel were cut out with a scalpel and destained in 30% (v/v) ACN (Merck: 1.00029.1000), 25 mM ammonium bicarbonate (Sigma: 09830) and incubated in an ultrasonic bath for 30 min. This step was repeated until all gel pieces appeared clear. Afterwards, the gel pieces were dehydrated using 100% (v/v) ACN for 15 min at room temperature. The acetonitrile was removed, and proteins were reduced and alkylated using 10 mM TCEP (Sigma: C4706) and 25 mM 2-chloroacetamide (CAA, Sigma: C0267) in 50 mM ammonium bicarbonate for 45 min at room temperature in the dark. Gel pieces were dehydrated afterwards as described above, and supernatant was removed. Trypsin/LysC (Promega: V5111) was resuspended in 100 mM ammonium bicarbonate to a concentration of 0.05 μg/μl, and 20 μl (= 1 μg) was added to each vial containing the dehydrated gel pieces. Acetonitrile was added to a final concentration of 10% (v/v), and protease digest was carried out overnight at 37°C and 300 rpm using a Thermomix (Eppendorf). Peptides were extracted in three steps: 1% (v/v) FA (Merck: 5.33002.0050); 60% (v/v) ACN, 1% (v/v) FA; and as last step 100% (v/v) ACN. Extraction was carried out for 15 min using an ultrasonic bath. After each step, the respective supernatant was removed, combined into a tube and freeze-dried (ScanVac Cool-Safe) overnight.

### C18 clean-up

StageTips were prepared in-house using C18 discs (Empore, Sigma: 66883), with a total of two C18 plugs per tip. Freeze-dried peptides were resuspended in TA2 (2% (v/v) ACN, 0.1% (v/v) TFA (Merck: 1.08262.0025)). StageTip was activated using 100% (v/v) acetonitrile and equilibrated using 0.1% (v/v) TFA. Peptides were bound to the C18 StageTip, desalted twice using 0.1% (v/v) TFA and eluted with 70% (v/v) ACN, 0.1% (v/v) TFA. Eluate was freeze-dried overnight.

### TMT labelling

TMT-10plex (Thermo Fisher: 90110) labels (0.8 mg) were resuspended in 100 μl anhydrous ACN (Sigma: 271004), vortexed and incubated for 5 min at room temperature. Freeze-dried eluted peptides were resuspended in 80 μl 100 mM TEAB (Thermo Fisher: 90114) and incubated in an ultrasonic bath for 15 min. Seventy microlitres of each sample was combined with 30 μl of the respective TMT10plex label (see Table EV2). TMT reaction was carried out at room temperature for 2 h. An aliquot of 10 μl of each sample was taken, and the reaction was stopped by addition of 50% (w/v) hydroxylamine (Sigma: 467804) to a final concentration of 5% (v/v). Stopped TMT reactions were combined and freeze-dried overnight.

### Mass spectrometry

Freeze-dried TMT-labelled peptides were resuspended in 20 μl TA2 and incubated in an ultrasonic bath for 15 min. Mass spectrometry was carried out on an Orbitrap Velos (Thermo Fisher) coupled to an Ultimate 3000 HPLC system (Thermo Fisher). One microlitre of peptides was injected and desalted for 5 min on an Acclaim PepMap 100 C18 column (5 μm particle size, 100 μm inner diameter, 2 cm length, 100 Å pore size, Thermo Fisher) using 3% (v/v) ACN, 0.1% (v/v) TFA at a flow rate of 5 μl/min. Afterwards, peptides were separated on an Acclaim PepMap 100 C18 column (2 μm particle size, 75 μm inner diameter, 50 cm length, 100 Å pore size, Thermo Fisher) using a two-buffer gradient system. Buffer component A was 0.1% (v/v) FA, Buffer component B consisted of 80% (v/v) ACN, 0.08% (v/v) FA. Peptides were eluted using a gradient from 3 to 35% (v/v) B over 101 min. This was followed by a gradient to 99% (v/v) B within 5 min and was held for 3 min. Afterwards, the column was washed for 5 min with 3% (v/v) B. Flow rate in all steps was 300 nl/min. Temperature of the column oven was set to 45°C. HPLC was coupled to the Orbitrap via an EASY-Spray ESI source (Thermo Fisher). Orbitrap acquired data in Nth order double play mode, scanning between $m/z$ 400 and 1,600 with the top 20 most intense $m/z$ features undergoing HCD fragmentation (minimum signal required: 2,000; isolation width: 2.0; normalized collision energy 35.0, activation time: 0.1). Internal calibration was performed using a lock mass at 445.120024 $m/z$.

### Data analysis

Raw files were searched using the Andromeda search engine integrated into MaxQuant (Cox *et al*, 2011; version 1.5.8.3) against a Uniprot human database downloaded on 08/11/2017 (20,239 sequences). Standard MaxQuant search parameters for Orbitrap data were used as follows: 4.5 ppm precursor tolerance with MS/MS tolerance of 20 ppm. Protease was specified as trypsin/P, fixed modification was set as carbamidomethylation of cysteine, and variable modifications were oxidation of methionine, protein N-terminal acetylation and phosphorylation of serine and threonine. Isobaric labelling was specified as TMT10plex, with omission of TMT-126 label. Reporter ion mass tolerance was set to 0.003 Da, and parent ion fraction (PIF) cut-off was set to 75%. Cut-offs for posterior error probabilities of peptide to spectrum matches, peptide and protein identifications were set to 1 per cent. MaxQuant data were analysed using an R-script with additional libraries ggplot2 (Wickham, 2009), reshape2 (Wickham, 2007) and VSN (Huber *et al*, 2002; Computer Code EV1; Appendix Fig S4). In brief, isotopically corrected reporter intensities were normalized and transformed using variance stabilizing normalization (VSN; Huber *et al*, 2002, 2003), and then, data were tested for differences in the mean of the groups using a *t*-test. Multiple testing was controlled for by lowering the significance threshold of $\alpha = 0.05$ to $\alpha = 0.00833$ [six *t*-tests, Bonferroni method (Computer Code EV1)].

## Measuring peptide kinase activity in CDKL5 immunoprecipitates

Cells were lysed in ice-cold wash buffer (50 mM HEPES (pH 7.5), 1% Triton X-100, 0.3 M sucrose, 300 mM NaCl) supplemented with 10 mM iodoacetamide, 10 ng/ml microcystin-LR, 2% (v/v) phosphatase inhibitor cocktail supplemented with a protease inhibitor cocktail (cOmplete™, EDTA-free Protease Inhibitor Cocktail) and 500 U/ml universal nuclease which were all added immediately before lysis. Cell debris was broken up using a needle and cleared by centrifugation at 20,800 *g* at 4°C. Extract protein concentration was quantified by using Pierce™ BCA Protein Assay Kit (Thermo Scientific 23225) in 96-well plates. For measuring kinase activity, extracts (1.0–2.5 mg) from HEK293 cells transiently expressing CDKL5-FLAG were incubated with 10 μl (settled) anti-FLAG agarose M2 affinity gel (Sigma-Aldrich) for 1 h. Beads were washed three times (15 min each) in wash buffer (without protease inhibitors) containing 1 M NaCl and then twice in kinase buffer (50 mM Tris 7.5, 10 mM MgCl$_2$, 0.1 mM EGTA). Beads were suspended in 15 μl kinase buffer containing 0.15 mM peptide substrate and 0.1% (v/v) 2-mercaptoethanol. Reactions were initiated with 5 μl of [γ-$^{32}$P]-ATP (final ATP concentration 0.1 mM) and incubated for 30 min at 30°C before being stopped by the addition of 10 μl of 0.1M EDTA. After centrifugation at 500 × *g*, supernatants (30 μl) were spotted onto P81-cation exchange (phosphocellulose) paper, and papers were washed in 75 mM orthophosphoric acid until background counts were at a minimum, dried in acetone and placed in scintillation counter to detect $^{32}$P incorporation by Cerenkov counting. The remaining beads were boiled in LDS sample buffer and subjected to SDS–PAGE followed by Western blotting to monitor CDKL5 abundance.

## Protein sequence alignments

Protein sequences from different species were obtained from Uniprot: http://www.uniprot.org/ or Protein Blast: https://blast.ncbi.nlm.nih.gov/Blast.cgi?PAGE=Proteins. Sequences to be aligned were introduced in Clustal Omega: https://www.ebi.ac.uk/Tools/msa/clustalo/, and the resulting alignments were then pasted in Boxshade server: https://embnet.vital-it.ch/software/BOX_form.html. Fifteen residues from each side of the RPXSA motif are shown for clarity in the alignments.

## Statistical analysis

All bar graphs and box plots in Figs 4–7 and EV2, EV3 and EV5 were generated using GraphPad Prism software using standard settings (version 5.0b, GraphPad Software, Inc.)

# Data availability

The global phosphoproteomics mass spectrometry data relating to Figs 1 and 2 have been deposited to the ProteomeXchange Consortium via the PRIDE partner repository (Jarnuczak & Vizcaino, 2017) with the dataset identifier PXD009374. The mass spectrometry proteomics data relating to the extracted ion chromatograms in Fig 5 have been deposited to the ProteomeXchange Consortium via the PRIDE partner repository (Jarnuczak & Vizcaino, 2017) with the dataset identifier PXD009327.

Expanded View for this article is available online.

## Acknowledgements

We thank the excellent technical support services of the MRC Protein Phosphorylation and Ubiquitylation Unit including the DNA Sequencing Service (coordinated by Gary Hunter), the MRC PPU tissue culture team (coordinated by Laura Fin) and the MRC PPU Reagents and Services teams (coordinated by Hilary McLauchlan and James Hastie). We thank David Campbell, Bob Gourlay, Axel Knebel and Dario Alessi for useful discussions. We are grateful to the Loulou Foundation for financial support of this work (IMM). This work was also supported by the Medical Research Council (grant number MC_UU_12016/1; MEM, JR) and the pharmaceutical companies supporting the Division of Signal Transduction Therapy Unit (Boehringer–Ingelheim, GlaxoSmithKline and Merck KGaA).

## Author contributions

JP and MT carried out the global phosphoproteomic mass spectrometry and relevant data analysis. IMM prepared the samples for the global phosphoproteomics experiment, and together with MEM carried out all of the other experiments in this study. FW carried out the extracted ion chromatogram analysis, and data analysis, shown in Fig 5 and wrote the MaxQuant data analysis R-script. MG helped with analysis of the activity of CDKL1-4. FCMB helped with antibody production and purification. RT and TM cloned all of the DNA constructs used in this study. JR conceived the project with input from IMM and MT and also wrote the paper.

## Conflict of interest

The authors declare that they have no conflict of interest.

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
