## [Review Process File · The EMBO Journal]

Phosphoproteomic screening identifies physiological substrates of the CDKL5 kinase

Ivan Muñoz, Michael Morgan, Julien Peltier, Florian Weiland, Mateusz Gregorczyk, Fiona Brown, Thomas J. Macartney, Rachel Toth, Matthias Trost & John Rouse.

Review timeline:

Submission date:	5 th April 2018
Editorial Decision:	7 th May 2018
Revision received:	14 th August 2018
Accepted:	10 th September 2018

Editor: Hartmut Vordermaier

Transaction Report:

1st Editorial Decision

7th May 2018

Thank you for submitting your manuscript on CDKL5 substrate screening for consideration by The EMBO Journal. It has now been assessed by three expert referees, and given their unanimously positive feedback, we shall be happy to publish it as a resource article, pending adequate revision of a number of specific queries raised in their reports. In this respect, I should point out that we would consider repetition of the analysis in neuronal cell types or patient cells beyond the scope of this first report on CDKL5 targets.

Please note that it is our policy to allow only a single round of experimental revision, making it important to diligently respond to all points raised at this stage. We generally allow three months as standard revision time, and it is our policy that competing manuscripts published during this period will have no negative impact on our final assessment of your revised study. Further information and guidelines on how to prepare a revision can be found below. In any case, please do not hesitate to contact me directly you should need any further clarifications or have any questions regarding your revision.

REFeree REPORTS.

Referee #1:

The paper by Munoz and colleagues is the first study to try to get to the heart of the substrate specificity and intracellular targets of the CDK/MAPK-like kinase CDKL5/STK9 (and by extension, some of the other CDKL protein kinases, which are also predicted to be Ser/Thr kinases). CDKL5 is relevant here because it is implicated in X and Y-linked diseases, and specific disease associated mutations are known and can be analysed.

The study is very well written, using a convincing scientific approach and is certainly original, uncovering as it does a couple of likely direct CDKL5 substrates including MAP1S and CEP131, which is particularly interesting based on its recently uncovered centriolar roles associated with genome stability. By exploiting knock-out and knock-in strategy, and aligning with impressive quantitative phosphoproteomics (the team is led by Rouse and Trost, internationally-acknowledged experts in this type of approach), several potential substrates, including CDKL5 were in U2OS cells.

I am strongly supportive of publication, largely because this paper reminds me of the types of thorough study that EMBO J used to publish in the 1990s and 2000s in this area when mechanistic analysis of protein kinases was in its infancy. Sadly, these types of study, carefully evaluating substrate specificity and toning down, but not ignoring, speculative cell biology in the first paper, are thin on the ground. In particular, no one has attempted a similar approach with CDKL5, to my knowledge, so this work is timely.

The length of the paper appears appropriate, and I only have a few comments that could be addressed by the authors:

Minor points:

p.5. ERK1 is picked as the example, is this not also true of ERK2, or am I missing something here?

Do the pathogenic CDKL5 mutations evaluated also impact on CDKL5 Tyr phosphorylation in the TEY motif of the activation segment? If so, this might be how they inactivate the kinase. In this context, there is also a Tyr and a Thr more C-terminal to the TEY motif but before the SPE (conserved in many kinases, actually). Is this phosphorylated in any of the MS studies? These phosphorylations can also impact on kinase activity, and might therefore be relevant in the case of CDKL5.

Figure S4 demonstrates differential expression of each CDKL5 when tagged with HA. No problems with this. But how do we know that these kinases don't actually phosphorylate MAP1S or CEP131? The authors are relying on the phosphoantibody revealing the activity of the kinase, rather than the reality in cells, which is that phosphatases might just dephosphorylate these substrates very quickly. SO I think the statement that they are not substrates need to be tempered, especially because in CDKL1 and CDKL2 (especially) there is clearly evidence of a weak signal. Probably better to state that CDKL5 appears to be more efficient under the conditions tested (asynchronous cell culture). To get to the bottom of this, have the IP's been tested using the MAP1S or CEP131-derived peptides in IP kinase assays? I couldn't find this in this context (analysis of CDKL1-5 proteins), but it is there in Figure 8, and provides convincing evidence in this context.

Do any of the author potential targets already known to be phosphorylated in cytoskeletal/cilia pathways (TPX2 et al) contain a potential R-P-X-S/T-A/G motif?

In Figure S3, why are only Ser residues considered for the substrate analysis. Is this due to the approach taken, or are there no Thr residues with a consensus (the 29 out of 194 Thr discussed on p.8) ? This might link conveniently to the peptide analysis showing Ser/Thr comparison in the phosphorylation site if discussed here. Also in this context, was Tyr tested in a synthetic peptide (perhaps of the activation segment itself?) as a potential phosphoacceptor, because an intriguing issue not covered by the current manuscript, is how Tyr might be incorporated into CDKL5; does it require an 'upstream' MAPKK like activity (ie a dual-specificity kinase) or is CDKL5 a dual-specificity kinase?! At its most simplistic, are any of the CDKL1-5 proteins tyrosine phosphorylated when immunoblotted with a pTyr antibody?

On p.26, to allow repetition by other authors, what is the concentration of the ATP in the assay, only a volume is given currently.

Re: MECP2 (in the context of RETT syndrome) and NGL-1 are mentioned in the Introduction, but then largely ignored. Can the authors conclude anything about the lack of appearance of these proteins in the cell work (e.g., would they be found in osteosarcoma cells as putative CDKL5 substrates?)

Referee #2:

Here the authors have characterized the understudied CDKL5 kinase, encoded by the CDKL5 gene on the X chromosome, in which it has recently been shown that mutations result in an early onset neurodegenerative disease defined as CDKL5 disease. They started by analyzing the total phosphoproteome of U2OS osteosarcoma cells lacking CDKL5 as a result of a CRISPR/Cas9 knockout compared to the CDKL5 KO cells re-expressing CDKL5 from a single locus under TET control, using TiO₂ phosphopeptide enrichment and TMT labeling for quantitative comparison. They identified 194 phosphopeptides that were upregulated >1.5 fold in the CDKL5-expressing cells and 159 that were decreased. Several of these sites were in CDKL5 itself, suggesting that they are autophosphorylation sites. Among the other sites dependent on CDKL5, they selected MAPS1 and CEP131, two proteins implicated in cytoskeleton/centrosome/primary cilia function, for further study, focusing on MAPS1 pS900 and CEP131 pS35 sites. They generated phosphospecific antibodies for MAPS1 pS900, and showed that this site was phosphorylated when FLAG-tagged MAPS1 was co-expressed with WT but not K42R kinase dead CDKL5 in 293 cells, using a S900A mutant MAPS1 as a control. Similar results were obtained with antibodies against pS35 CEP131. They noted that both sites have an RPXSA consensus, and using short synthetic peptides corresponding to the MAPS1 S900 sites with substitutions at different residues around the target Ser as substrates for purified FLAG-tagged CDKL5 in vitro they showed that the upstream Arg and Pro and the downstream Ala were important determinants of peptide phosphorylation efficiency, and deduced that both MAPS1 S900 and CEP131 S35 are direct substrates for CDKL5. They also identified S1115 in DLG5, a protein involved in signaling to the MT-based cytoskeleton, as another substrate of CDKL5 using the same approaches. Next, they showed that Y171 in a TEY motif in the CDKL5 catalytic domain activation loop was important for MAPS1/CEP131 phosphorylation activity in vivo, and in vitro against MAPS1 S900 and CEP131 S35 peptides, whereas T169 in the TEY motif was not required for CDKL5 activity. Finally, they analyzed the effects of a set of known CDKL5 disease point mutations on catalytic activity in vivo with co-transfected MAPS1 or CEP31 or in vitro against synthetic peptide substrates, and found that G20D, L64P, I72T, R178W and Q219P, lying in the catalytic domain, all abolished activity, whereas six other CDKL5 point mutants downstream of the catalytic domain had WT activity.

These carefully done phosphoproteomic studies have led to identification of what appear to be the first authentic substrates for the CDKL5 kinase, and shown that "CDKL5 disease" mutants are loss of function (as one might have predicted). The identification of these first authentic CDKL5 substrates could ultimately lead to a deeper understanding of why CDKL5 loss of function mutations cause CDKL5 neurodevelopmental disease.

General point: CDKL5 is normally expressed in neurons, which are presumably the causal cell in CDKL5 disease, but also in other cell types. It would be desirable to analyze for CDKL5 substrates in a neuronal cell type, possibly patient iPSC-derived neurons compared to WT iPSC-derived neurons (or CDKL5 KO iPSC-derived neurons). Another issue that was not discussed is where CDKL5 is localized in cells; it has an NLS and an NES - does it shuttle between nucleus and cytoplasm. In this regard, would CDKL5 be expected to have access to proteins in the primary cilium?

Points: 1. The Introduction is rather long, and the section on putative CDKL5 substrates could be shortened considerably.

2. Figure 1: U2OS cells are female and are reportedly hypertriploid with many marker chromosomes. Since these cells are not diploid, they may contain more than two CDKL5 alleles, and all these would have to be disrupted by CRISPR/Cas9 for a complete knockout. The discussion of the sequence analysis of the U2OS KO clones suggest that the authors believe that there are only two CDKL5 alleles. However, it is possible that there is a third CDKL5 allele and that the weak CDKL5 antibody positive band running at the same size as CDKL5 is in fact due to a low level of CDKL5 being expressed from a third CDKL5 allele? What does analysis of CDKL5 RNA levels in the KO cells show?

3. Figure 1C: Why didn't the level of exogenous CDKL5 increase with increasing TET dose?
4. By using TiO₂ enrichment the authors only looked at pS/pT sites in these cells, although apparently pY171 was subsequently identified. Did they ever identify a pT169EpY171 peptide - the tryptic peptide is relatively short and should be detectable by MS.
5. As the authors indicate, all of the phosphorylation sites found in the C-terminal region of CDKL5 are in SP motifs and none have an RPXSA consensus. However, RPISS (S959) in the CDKL5 C-tail is in an RPXS motif, but was not reported to be phosphorylated. Are these sites also observed when K42R kinase dead CDKL5 is expressed in the CDKL5 KO cells, or do they require CDKL5 to be active?
6. Figure 5: For the peptide phosphorylation assays, the authors should determine whether the effects of the peptide "mutations" on phosphorylation efficiency are a result of differences in K_m or V_{max} or both.
7. What is the upstream activator kinase for CDKL5 in U2OS cells? Is Y171 an autophosphorylation site? Would T169 be phosphorylated in a more appropriate cell type, and possibly play a functional role?
8. Did the authors compare the level of phosphorylation of MAPS1 S900 and CEP131 S35 in the WT U2OS cells versus CDKL5 KO cells?
9. How specific are the pS900 and pS35 antibodies when used against a whole cell lysate. Do the anti-pS900 antibodies crossreact with pS35 CEP131 in an IP/IB experiment or with a pS35 phosphopeptide or vice versa - after all, the RPXpSA residues are the same for both sites.
10. Figure 7: The deleterious effect of the Y171A mutation on CDKL5 activity indicates that Y171 is important, but not necessarily that it has to be phosphorylated for activity. Although this seems likely, a more conservative Y171F mutation should have been tested.
11. Can CDKL5 be co-precipitated with MAPS1 and CEP131 when co-expressed, i.e. is there a CDKL5 substrate docking mechanism?
13. To establish that MAPS1 S900 and CEP131 S35 are direct substrates for CDKL5, ideally full length proteins rather than short peptides need to be tested.
14. Figure 8: Is there a structural model of the CDKL5 catalytic domain that could be used to explain why these mutations are deleterious to kinase activity?

Referee #3:

Review comments:

In the manuscript titled 'Phosphoproteomic screening identifies cytoskeleton regulators as novel targets of the CDKL5 kinase', Munoz and co-workers characterized a novel kinase named CDKL5. Using CDKL5 knockout and CDKL5 re-introduced cell models, and multiplexed phosphoproteomic screening approach, the authors identified candidate substrate proteins MAP1S and CEP131. Further follow-up studies were carried out to generate phospho-site specific antibodies for MAP1S and CEP131, identify putative CDKL5 substrate motif, elucidate CDKL5 activity regulation through TEY motif, and categorize the impact of many clinical-relevant CDKL5 mutations. Overall the quality of data is great and characterization on a novel kinase is thorough. Given the rising importance of CDKL5 in neurological diseases, conclusions presented in this manuscript will be an important addition to the community knowledge base. With a few questions to be addressed, this manuscript can be suitable for publication in EMBO.

Major:

1. The phosphoproteomic data lacks of protein level data for appropriate interpretation on the site-

level changes. For instance the most changing sites were CDKL5 itself, which is likely due to CDKL5 protein level difference in the two cell models (KO vs. re-expressed), rather than PTM regulation. It would be important to confirm MAP1S and CEP131 protein level in these two cell lines, either by MS or western blot.

2. In order to conclude CDKL1-4 can't phosphorylate MAP1S and CEP131 in cells, the authors need to confirm expressed CDKL1-4 are in the active form. Because the results were negative, it's not clear this is due to inactive kinases, or incorrect substrates.

3. In the motif and gene ontology analyses, what was used as background dataset? How is SP motif enrichment in the subset that showing >1.5-fold, versus overall phospho dataset?

4. In validating the importance of TEY motif for CDKL5 kinase activity, Y->A mutation abolishes kinase activity. It would be important to do phospho-mimetic mutation Y->D to check if phosphorylation is essential.

Minor:

5. Page 22 method section: AGC target of 2000 ions for Orbitrap MS2 is pretty low to me. Can the authors confirm this number?

6. Page 22, 25: Database search parameters were missing variable or fixed TMT modification on K and n-term.

7. Figure 8A 'hi' and 'lo' labels were mislabeled. Need to be swapped.

Suggestion:

8. Figure 3 and 4 were same analyses applied to two proteins. Would the authors consider combining into one figure as multiple panels to reduce total number of figures? Same thought also applies to figure 7 and 8.

Referee #1:

Minor points:

1. p5. ERK1 is picked as the example, is this not also true of ERK2, or am I missing something here?

The reviewer is correct; we've amended the text to include ERK2.

2. Do the pathogenic CDKL5 mutations evaluated also impact on CDKL5 Tyr phosphorylation in the TEY motif of the activation segment?

We have addressed this point. We found that CDKL5, but not a kinase-dead (K42R) version, cross-reacts with 4G10 anti-phosphotyrosine antibodies (before and after incubating immunoprecipitates with Mg²⁺-ATP) (Fig. 6A). Therefore, CDKL5 autophosphorylates on tyrosine. Mutating Y171 in the T-loop but not T169 (or Y168) also abolished CDKL5 Tyr phosphorylation (Fig. 6A). In these experiments, we were not sure that CDKL5 Tyr phosphorylation occurs in the T-loop so we raised an antibody against phospho-Y171. We found that CDKL5, but not the K42R kinase-dead mutant, cross-reacts with these antibodies and mutating Y171 but not T169 (or Y168) prevents cross-reactivity (Fig. 6B). These data show, therefore, that CDKL5 autophosphorylates on Y171; the data in Figs. 6E, F and EV5 show that Y171 is important for CDKL5 activity. Interestingly, CDKL5 autophosphorylation is likely to be intramolecular as CDKL5 cannot phosphorylate synthetic T-loop peptides (Fig. 6C).

We found that the pathogenic mutations G20D and L64P which abolish CDKL5 activity towards MAP1S and CEP131 cause a major reduction in tyrosine phosphorylation of CDKL5 where the non-pathogenic variants L302F, V718M and V999M do not (Fig. 6C). Surprisingly, the pathogenic mutation R178W which we showed in Fig 7 abolishes CDKL5 activity towards MAP1S and CEP131 has no apparent effect on CDKL5 tyrosine autophosphorylation (Fig. EV6). It may be that R178W is required for contacting exogenous substrates which would explain why it is required for MAP1S and CEP131 phosphorylation but not for phosphorylation of CDKL5 itself. More work will be needed to address this point.

In this context, there is also a Tyr and a Thr more C-terminal to the TEY motif but before the SPE. Is this phosphorylated in any of the MS studies?

We found no evidence in our *in vivo* CDKL5 phospho-site mapping studies for phosphorylation of these sites.

3. Figure S4 demonstrates differential expression of each CDKL5 when tagged with HA. No problems with this. But how do we know that these kinases don't actually phosphorylate MAP1S or CEP131? The authors are relying on the phosphoantibody revealing the activity of the kinase, rather than the reality in cells, which is that phosphatases might just dephosphorylate these substrates very quickly. SO I think the statement that they are not substrates need to be tempered, especially because in CDKL1 and CDKL2 (especially) there is clearly evidence of a weak signal.

Probably better to state that CDKL5 appears to be more efficient under the conditions tested (asynchronous cell culture). To get to the bottom of this, have the IP's been tested using the MAP1S or CEP131-derived peptides in IP kinase assays. We investigated CDKL1-4 activity in other ways. For example, we tested the ability of CDKL1-5 to autophosphorylate after incubation of FLAG precipitates with

radiolabelled ATP. We observed phosphorylation of CDKL1-4 under these conditions but in each case the activity was also seen with kinase-dead mutants. We think this is explained by contaminating kinases in the precipitates. We also tested the ability of FLAG-CDKL1-4 to autophosphorylate using anti-phosphotyrosine 4G10 antibodies. We saw that CDKL1 is phosphorylated on tyrosine but the signal is unaffected by a kinase-inactivating mutation. So autophosphorylation did not indicate whether CDKL1-4 are active.

We tested the activity of CDKL1-5 (wild-type or kinase-dead in each case) towards the MAP1S peptide as requested. We find that CDKL2 can phosphorylate the MAP1S peptide at around 50% the level of CDKL5, but no significant activity was seen with CDKL1,3 or 4 (Fig. EV2D). The text has been amended accordingly to mention this point, and our statements along these lines have been tempered. We stress that CDKL1,3 and 4 may simply be inactive under the conditions used in this study, but perhaps their activity (and perhaps the activity of all CDKLs) is regulated in response to extracellular or intracellular stimuli.

4. Do any of the author potential targets already known to be phosphorylated in cytoskeletal/cilia pathways (TPX2 et al) contain a potential R-P-X-S/T-A/G motif?

In our phospho-proteomics dataset are 3 further proteins whose GO terms include either “Cili(um/a)”, “Cytoskel(leton/letal)” or “mitotic spindle” and contain a RPX[S,T][A,G] motif. These are ARAP3_HUMAN (RPTTG), DOCK7_HUMAN (RPITA) and ARHGEF17_HUMAN (RPTTA and RPLTG). We aligned these proteins to the paralogues from *Pan troglodytes*, *Bos Taurus*, *Equus caballus*, *Mus musculus*, *Rattus norvegicus*, *Gallus gallus*, *Danio rerio* and *Xenopus tropicalis*.

In ARAP3, the relevant threonine of the RPTTG motif was conserved down to *Rattus norvegicus* (as in order of the species mentioned above, with omission of *Pan troglodytes* as no paralogue could be identified). This threonine was exchanged for a serine in *Gallus gallus*, with a further exchange of [A,G] to a lysine (RPASK). In ARHGEF17, the second threonine in the RPTTA motif was only conserved in *Pan troglodytes*. The threonine of the second RPX[S,T][A,G] like motif (RPLTG) was as well only conserved in *Pan troglodytes*, with an exchange to proline in *Bos Taurus*. *Mus musculus* and *Rattus norvegicus* exhibited a threonine to serine exchange in this motif. In DOCK7, the RPITA motif was conserved in all aligned species, with omission of *Equus caballus* and *Gallus gallus* due to missing paralogues.

We did not detect any phospho-peptides containing the RPX[S,T][A,G] motif sequences from any of the above proteins in our phospho-proteomics screen. Thr-189 of ARHGEF17 (RPLTG) does appear in the Phosphosite database, but no validation has been reported. Based on the above considerations, we have not mentioned ARAP3, DOCK7 or ARHGEF17 in the text.

5. In Figure S3, why are only Ser residues considered for the substrate analysis. Is this due to the approach taken, or are there no Thr residues with a consensus (the 29 out of 194 Thr discussed on p.8)?

Due to the low number of Thr-phosphorylated peptides compared to Ser-phosphorylated peptides in our screen, far fewer of the motifs that emerged are statistically significant. Apart from the TP motif, no Thr phospho-motifs were significantly enriched. We have added the TP motif to Appendix Figure S3.

This might link conveniently to the peptide analysis showing Ser/Thr comparison in the phosphorylation site if discussed here. Also in this context, was Tyr tested in a synthetic peptide (perhaps of the activation segment itself?) as a potential phosphoacceptor, because an intriguing issue not covered by the current manuscript, is how Tyr might be incorporated into CDKL5; does it require an 'upstream' MAPKK like activity (ie a dual-specificity kinase) or is CDKL5 a dual-specificity kinase?!

As described in the response to point 2 above we found that CDKL5 autophosphorylates on Y171 in the T-loop. However FLAG-CDKL5 immunoprecipitates have no detectable activity against peptides corresponding to the CDKL5 T-loop under conditions where the MAP1S S⁹⁰⁰ peptide is phosphorylated robustly. Furthermore, substituting S⁹⁰⁰ for a Tyr residue in the MAP1S peptide resulted in complete loss of MAP1S peptide phosphorylation by CDKL5. Thus, although CDKL5 can autophosphorylate on tyrosine, it cannot phosphorylate Tyr-containing peptides in trans, and can only phosphorylate exogenous substrates on Ser/Thr residues.

At its most simplistic, are any of the CDKL1-5 proteins tyrosine phosphorylated when immunoblotted with a pTyr antibody?

We now show that CDKL5 autophosphorylates on Y171 in Fig 6. We also tested Tyr phosphorylation of FLAG-CDKL1-4 using 4G10 antibodies. We saw that only CDKL1 is phosphorylated on tyrosine but the signal is unaffected by a kinase-inactivating mutation. We have not shown these data as they're not so informative.

On p.26, to allow repetition by other authors, what is the concentration of the ATP in the assay, only a volume is given currently.

This is an unfortunate omission, which has now been corrected. The final ATP concentration in the assay is 0.1mM.

Re: MECP2 (in the context of RETT syndrome) and NGL-1 are mentioned in the Introduction, but then largely ignored. Can the authors conclude anything about the lack of appearance of these proteins in the cell work (e.g., would they be found in osteosarcoma cells as putative CDKL5 substrates?)

We have tested expression of MECP2 and NGL-1 in U2OS cells by western blotting which showed that they are expressed at easily detectable levels. On that basis one would expect that lack of expression would not account for our failure to detect CDKL5-dependent phosphorylation of these proteins in our screen. It would be interesting to compare the ability of CDKL5 to phosphorylate MECP2 and NGL-1 in parallel with MAP1S/CEP131/DLG5. We have been unable to express full-length versions of these proteins however and so we have not done this experiment.

Referee #2:

General point: CDKL5 is normally expressed in neurons, which are presumably the causal cell in CDKL5 disease, but also in other cell types. It would be desirable to analyze for CDKL5 substrates in a neuronal cell type, possibly patient iPSC-derived neurons compared to WT iPSC-derived neurons (or CDKL5 KO iPSC-derived neurons).

This is an excellent point and one that we are addressing in detail. However, in agreement with editorial advice, we feel that repetition of the analysis in neuronal cell types or patient cells is beyond the scope of this first report on CDKL5 targets.

Another issue that was not discussed is where CDKL5 is localized in cells; it has an NLS and an NES - does it shuttle between nucleus and cytoplasm. In this regard, would CDKL5 be expected to have access to proteins in the primary cilium?

GFP-tagged CDKL5 expressed transiently or stably is located in both cytoplasm and nucleus in different human tissue culture cell lines in our hands; this is what has been observed in a range of reports in the literature. Using antibodies we raised against mouse CDKL5, we found that CDKL5 is located in both cytoplasm and nucleus in primary cortical neurons from mice; staining was abolished in primary neurons from CDKL5 KO mice. Therefore, CDKL5 would be expected to have access to the primary cilium. On this note, a recent report showed that CDKL5 localised to the base of primary cilia in human cells (PMID: 29420175) which we cite in the discussion.

Points: 1. The Introduction is rather long, and the section on putative CDKL5 substrates could be shortened considerably.

We have now shortened the Introduction substantially.

2. Figure 1: U2OS cells are female and are reportedly hypertriploid with many marker chromosomes. Since these cells are not diploid, they may contain more than two CDKL5 alleles, and all these would have to be disrupted by CRISPR/Cas9 for a complete knockout. The discussion of the sequence analysis of the U2OS KO clones suggest that the authors believe that there are only two CDKL5 alleles. However, it is possible that there is a third CDKL5 allele and that the weak CDKL5 antibody positive band running at the same size as CDKL5 is in fact due to a low level of CDKL5 being expressed from a third CDKL5 allele? What does analysis of CDKL5 RNA levels in the KO cells show?

The text here was confusing, we apologize for this. In clone 7 shown in Appendix Fig. S1 we found no wild type CDKL5 alleles either by genomic sequencing or by RT-PCR, but instead we found two classes of disrupted CDKL5 alleles. In clone 13, we found no wild type CDKL5 alleles either by genomic sequencing or by RT-PCR and only one class of disrupted CDKL5 allele – that is, all disrupted alleles had the same kind of alteration. We have amended the text to make this point and avoid confusion.

3. Figure 1C: Why didn't the level of exogenous CDKL5 increase with increasing TET dose?

In other stable cell lines we've generated in the U2OS FlpIn TREX system, these concentrations show increasing protein expression with increasing Tet dose. In the case of CDKL5, expression appears to be maximal at the lowest doses of Tet used.

4. By using TiO₂ enrichment the authors only looked at pS/pT sites in these cells, although apparently pY171 was subsequently identified. Did they ever identify a pT169EpY171 peptide - the tryptic peptide is relatively short and should be detectable by MS.

In our experience, pTyr sites are also enriched by TiO₂ chromatography. None of the pTyr sites changed in abundance in our screen, and the T-loop phosphosites were not detected in our large-scale proteomics screen even though by western blotting we found CDKL5 to be Tyr-phosphorylated at Y171. Several years ago we did a targeted phospho-peptide SRM analysis of MAP kinase T-loop peptides which are very similar to the T-loop site of CDKL5 (unpublished). These doubly-phosphorylated T-loop peptides do not ionise well and are rather difficult to detect in complex samples, probably explaining why we did not identify them in the phosphoproteomics screen. An alternative explanation might be a low stoichiometry of Y171 phosphorylation in cells, perhaps because CDKL5 is not activated until exposure to a stimulus. Perhaps the active pool in our precipitates is the cilium-localized pool.

5. As the authors indicate, all of the phosphorylation sites found in the C-terminal region of CDKL5 are in SP motifs and none have an RPXSA consensus. However, RPISS (S959) in the CDKL5 C-tail is in an RPXS motif, but was not reported to be phosphorylated. Are these sites also observed when K42R kinase dead CDKL5 is expressed in the CDKL5 KO cells, or do they require CDKL5 to be active?

The SP phospho-sites in CDKL5 that we identified in the phospho-proteomic screen are still phosphorylated in the kinase-dead version (data not shown) and therefore must be phosphorylated by Pro-directed kinases. We have not looked into this further, however, as mutating these sites had no detectable effect on CDKL5 activity in vitro or in cells (data not shown).

6. Figure 5: For the peptide phosphorylation assays, the authors should determine whether the effects of the peptide "mutations" on phosphorylation efficiency are a result of differences in Km or Vmax or both.

We agree that it's important to determine how pathogenic mutations exert their inhibitory effects on CDKL5, and looking at reaction kinetics would be one way to do this. However, because we have been unable to express active full length CDKL5 in recombinant form we have had to rely on FLAG-CDKL5 immunoprecipitates to assay kinase activity. In our experience, it is difficult to get reliable kinetic parameters from experiments done with immunoprecipitates. We have cloned many different orthologues of CDKL5 in the hope that at least one of them may express at high enough yield for us to carry out detailed kinetic analyses.

7. What is the upstream activator kinase for CDKL5 in U2OS cells? Is Y171 an autophosphorylation site?

We have found that autophosphorylation of CDKL5 on Y171 in the T-loop is critical for activity. We have addressed this point. We found that CDKL5, but not a kinase-dead (K42R) version, cross-reacts with 4G10 anti-phosphotyrosine antibodies (before and after incubating immunoprecipitates with Mg²⁺-ATP) (Fig. 6A). Therefore, CDKL5 autophosphorylates on tyrosine. Mutating Y171 in the T-loop but not T169 (or Y168) also abolished CDKL5 Tyr phosphorylation (Fig. 6A). In these experiments, we are not sure that CDKL5 Tyr phosphorylation occurs in the T-loop so we raised an antibody against phospho-Y171. We found that CDKL5, but not the K42R kinase-dead mutant, cross-reacts with these antibodies and mutating Y171 but

not T169 (or Y168) prevents cross-reactivity (Fig. 6B). These data show, therefore, that CDKL5 autophosphorylates on Y171; the data in Figs. 6E, F and EV5 show that Y171 is important for CDKL5 activity. Interestingly, CDKL5 autophosphorylation is likely to be intramolecular as CDKL5 cannot phosphorylate synthetic T-loop peptides (Fig. 6C).

We do not yet know if phosphorylation of the T-loop is regulated – perhaps by phosphorylation at other sites by an upstream kinase, for example. This will be investigated in the future. Interestingly, the SP phospho-sites in CDKL5 that we identified in the phospho-proteomic screen are still phosphorylated in the kinase-dead version (data not shown) and therefore must be phosphorylated by Pro-directed kinases. We have not looked into this further, however, as mutating these sites had no detectable effect on CDKL5 activity in vitro or in cells (data not shown).

Would T169 be phosphorylated in a more appropriate cell type, and possibly play a functional role?

Perhaps, but it's not clear at the moment. It's possible that T169 is phosphorylated in response to extracellular or intracellular stimuli and this will be tested in the future.

8. Did the authors compare the level of phosphorylation of MAPS1 S900 and CEP131 S35 in the WT U2OS cells versus CDKL5 KO cells?

Endogenous CDKL5, MAP1S and CEP131 all appear to be expressed at low abundance in U2OS cells. On top of that, the antibodies we raised do not appear to be sensitive enough to recognise the phosphorylated forms of endogenous MAP1S and CEP131 despite numerous attempts. A recent screen carried out by my colleague Dario Alessi in collaboration with Matthias Mann identified Rab proteins as key substrates of the LRRK2 kinase mutated in Parkinson's disease. Sheep polyclonal antibodies raised against the Rab phospho-sites allowed validation of LRRK2-dependent Rab phosphorylation in cells, as we have done with CDKL5 and MAP1S/CEP131. However, the analysis of endogenous Rab phosphorylation in biologically meaningful ways required rabbit monoclonal antibody programmes. We are very excited that we recently obtained funding to generate rabbit monoclonal antibodies against CDKL5, phospho-MAP1S and phospho-CEP131. These reagents will hopefully allow rigorous analysis of endogenous MAP1S and CEP131 in various organs and cell types in the future.

9. How specific are the pS900 and pS35 antibodies when used against a whole cell lysate. Do the anti-pS900 antibodies crossreact with pS35 CEP131 in an IP/IB experiment or with a pS35 phosphopeptide or vice versa - after all, the RPXpSA residues are the same for both sites.

Since MAP1S and CEP131 are low abundance in cells their phosphorylation cannot be detected by blotting whole cell extracts. Therefore, we analyse FLAG-IPs from cells expressing FLAG-CEP131 or MAP1S.

The MAP1S pS900 and CEP131 pS35 antibodies were raised against 15-amino acid peptides encompassing the relevant RPXSA motif, and therefore it is likely that residues outside RPXSA motif formed part of the epitope. Consistent with this idea, the anti-pS⁹⁰⁰ antibodies show almost no cross-reactivity with pS³⁵ CEP131 and the anti-pS³⁵ antibodies show almost no cross-reactivity with pS⁹⁰⁰ MAP1S. As mutating S⁹⁰⁰ to Ala prevented the anti-pS⁹⁰⁰ antibodies from recognizing anti-MAP1S

precipitates, we know that it is MAP1S the antibodies cross react with in the FLAG precipitates analysed in this study. The same kind of control was done for pS³⁵ CEP131 antibodies, and so we're sure of the specificity of these antibodies too.

10. Figure 7: The deleterious effect of the Y171A mutation on CDK5L activity indicates that Y171 is important, but not necessarily that it has to be phosphorylated for activity. Although this seems likely, a more conservative Y171F mutation should have been tested.

As mentioned above, we have now shown CDKL5 autophosphorylates on Y171 using phospho-Y171 antibodies. We tested a Y171F mutant and it shows the same reduction in CDKL5 activity as the Y171A mutant (mentioned as data not shown).

11. Can CDKL5 be co-precipitated with MAPS1 and CEP131 when co-expressed, i.e. is there a CDKL5 substrate docking mechanism?

We have not been able to detect a convincing interaction between CDKL5 and its substrates in cells – with or without overexpression of either kinase or substrates or both. In our experience, enzyme-substrate interactions can be hard to detect perhaps because they are transient.

13. To establish that MAPS1 S900 and CEP131 S35 are direct substrates for CDKL5, ideally full length proteins rather than short peptides need to be tested.

We have been trying for some time to address this point, but we have been unable to express full length human or mouse MAP1S or CEP131. We will have to resort to trying orthologues from other species but this is beyond the scope the present study.

14. Figure 8: Is there a structural model of the CDKL5 catalytic domain that could be used to explain why these mutations are deleterious to kinase activity?

Unfortunately there is not a convincing explanation, and modelling CDKL5 on to other kinase structures has not been informative.

Referee #3:

Review comments:

1. The phosphoproteomic data lacks of protein level data for appropriate interpretation on the site-level changes. For instance the most changing sites were CDKL5 itself, which is likely due to CDKL5 protein level difference in the two cell models (KO vs. re-expressed), rather than PTM regulation. It would be important to confirm MAP1S and CEP131 protein level in these two cell lines, either by MS or western blot.

We have confirmed by western blotting that MAP1S and CEP131 are expressed at similar levels in CDKL5 KO cells and the parental cells.

2. In order to conclude CDKL1-4 can't phosphorylate MAP1S and CEP131 in cells, the authors need to confirm expressed CDKL1-4 are in the active form. Because the results were negative, it's not clear this is due to inactive kinases, or incorrect substrates.

We have tried to address this point in several ways. First, we tested the ability of CDKL1-5 to autophosphorylate after incubation of FLAG precipitates with radiolabelled ATP. We observed phosphorylation of CDKL1-4 under these conditions

but in each case the activity was also seen with kinase-dead mutants. We think this is explained by contaminating kinases in the precipitates. We also tested the ability of FLAG-CDKL1-4 to autophosphorylate using anti-phosphotyrosine 4G10 antibodies. We saw that CDKL1 is phosphorylated on tyrosine but the signal is unaffected by a kinase-inactivating mutation. So autophosphorylation did not indicate whether CDKL1-4 are active.

We tested the activity of CDKL1-5 (wild-type or kinase-dead in each case) towards the MAP1S peptide. We find that CDKL2 can phosphorylate the MAP1S peptide at around 50% the level of CDKL5, but no significant activity was seen with CDKL1,3 or 4. The text has been amended accordingly to mention these points. We stress that CDKL1,3 and 4 may simply be inactive under the conditions used in this study, but perhaps their activity (and perhaps the activity of all CDKLs) is regulated in response to extracellular or intracellular stimuli.

3. In the motif and gene ontology analyses, what was used as background dataset? How is SP motif enrichment in the subset that showing >1.5-fold, versus overall phospho dataset?

For the motif-x analysis the phosphosites significantly more abundant in CDKL5-expressing cells were compared to the whole human proteome as background dataset, which showed an enrichment of the SP motif. When the more abundant sites are compared to the overall dataset, there is no enrichment for SP anymore. As the overall number of changing phosphopeptides was relatively small (165 Ser sites, compared to >12,000 phosphosites), enrichment analysis was not very sensitive statistically. We therefore focussed particularly on the high-changing peptides and as the RxxS motif was present in the top two peptides (from CEP131 and MAP1S) that were not from CDKL5 itself, we focussed on these.

4. In validating the importance of TEY motif for CDKL5 kinase activity, Y->A mutation abolishes kinase activity. It would be important to do phospho-mimetic mutation Y->D to check if phosphorylation is essential.

In our hands phospho-mimic D/E substitutions frequently fail to mimic phosphorylation, which makes this type of experiment difficult to interpret.

Minor:

5. Page 22 method section: AGC target of 2000 ions for Orbitrap MS2 is pretty low to me. Can the authors confirm this number?

AGC target for CID is indeed 2000 ions (which is close to the default parameter of the universal method, which is 3000 ions). In our experience the sensitivity of the ion trap in CID is so high that 2000 ions are sufficient.

6. Page 22, 25: Database search parameters were missing variable or fixed TMT modification on K and n-term.

We have added the missing database search parameters in the method section.

7. Figure 8A 'hi' and 'lo' labels were mislabeled. Need to be swapped.

The figure has been amended.

Suggestion:

8. Figure 3 and 4 were same analyses applied to two proteins. Would the authors

consider combining into one figure as multiple panels to reduce total number of figures? Same thought also applies to figure 7 and 8.

We have combined Figures 3 and 4 in the revised manuscript. We prefer not to combine Figures 7 and 8.

Accepted

10th September 2018

Thank you for submitting your final revised manuscript for our consideration. I am pleased to inform you that following re-assessment by the original referee 1 (see comments below), we have now accepted it for publication in The EMBO Journal!

REFeree REPORTS.

Referee #1:

My comments after reading the response to all reviewers, including my own, are that the manuscript is now suitable for publication in the EMBO J. All points have been answered, and some of the data has been improved or merged, which in both cases helps the paper.

Corresponding Author Name: John Rouse
Journal Submitted to: EMBO Journal
Manuscript Number: EMBOJ-2018-99559R